# ENFORCING LATENT EUCLIDEAN GEOMETRY IN VAES FOR STATISTICAL MANIFOLD INTERPOLATION

## ABSTRACT

Latent linear interpolations are a powerful tool for navigating the representation space of deep generative models. This aspect is particularly relevant in applied settings, where meaningful latent traversals can be learnt to represent the evolution of a system's trajectory and mapped back to the often noisy and high-dimensional data space. However, when real data lies on a manifold with non-trivial geometry, linear interpolations of the representation space do not directly correspond to geodesic paths along the manifold unless enforced. An example of such a setting is scRNA-seq, where high-dimensional and discrete cellular data is assumed to lie on a negative binomial statistical manifold modelled by the decoder of a variational autoencoder. We introduce FlatVI, a novel training framework enforcing Euclidean geometry in the latent space of discrete-likelihood variational autoencoders modelling count data. In our modelling setting, straight lines in the latent domain are regularised to approximate geodesic interpolations in the decoded space, improving the combination of our model with methods assuming Euclidean latent geometry. Results on simulated data empirically support our claims, while experiments on temporally resolved biological datasets show improvements in the reconstruction of cellular trajectories and the learning of biologically meaningful velocity fields.

## 1 INTRODUCTION

Generative models for representation learning like Variational Autoencoders (VAEs) have influenced computational sciences in multiple fields (Zhong & Meidani, 2023; Lopez et al., 2018; Griffiths & Hernández-Lobato, 2020). One reason is that real-world experimental data often poses significant modelling challenges as it is inherently noisy, high-dimensional, and complex (Sarker, 2021). As a solution, learning a low-dimensional and dense latent representation of the data has gained traction in applied machine learning. One example is reconstructing population dynamics, where the evolution of individual particles in complex systems is learnt from disjoint samples of observations collected at subsequent time points. In the presence of high-dimensional systems, modelling interpolations in a well-behaved low-dimensional data embedding can offer valuable insight into the system's evolution through time (Džeroski & Todorovski, 2003; Bunne et al., 2022). Furthermore, with the decoder parameterising a flexible likelihood model, VAEs present a promising avenue for representing both continuous and discrete data. The latter setting has demonstrated unprecedented potential in cellular data, particularly in gene expression, which is measured in counts that reflect the number of RNA molecules produced by thousands of genes and is collected through single-cell RNA sequencing (scRNA-seq). (Haque et al., 2017). Such a technique allows the measurement of thousands of genes in parallel and the resulting discrete count vectors describe the state of a cell across diverse biological settings (Regev et al., 2017).

The representation learnt by VAEs is tightly connected to Riemanniann geometry, as one can see the latent space as a parametrisation of a low-dimensional manifold (Arvanitidis et al., 2020). When it comes to modelling latent population dynamics, popular approaches still rely on assuming Euclidean geometry in the representation space (Bunne et al., 2022; Tong et al., 2020; 2023). A notable example is the standard formulation of dynamic Optimal Transport (OT) (Benamou & Brenier, 2000), which exploits straight-line interpolations between observations for learning a time-dependent map across population snapshots. *However, building latent trajectories upon the Euclidean assumption is suboptimal when the data lies on a non-Euclidean manifold*, as straight latent lines do not necessarily reflect geodesic paths on the manifold spanned by the decoder, see Fig. 1.

To learn effective interpolations on real data manifolds, we establish the following desiderata: **(i)** Approximate trajectories on intractable data manifolds via interpolations on a simpler latent manifold with tractable geometry. **(ii)** Design a VAE decoding scheme that maps the shortest paths on the latent manifold to geodesics in the decoded space. **(iii)** Formalise the geodesic matching framework in a way that supports a flexible choice of the decoder's likelihood. To achieve **(i)** and **(ii)**, existing methods regularise the latent representation of Gaussian AEs using Euclidean geometry (Chen et al., 2020; Yonghyeon et al., 2021), but limit their application to continuous data by neglecting the decoder's general likelihood model support. Other works explore the connection between stochastic decoders' geometry and the latent space manifold (Arvanitidis et al., 2021), even for discrete data, but do not address regularising the latent manifold to a simple, traversable geometry while preserving geodesic paths on the decoded manifold. To our knowledge, no VAE-based method integrates all desiderata.

In this work, we close this gap and introduce FlatVI, Flat Variational Inference, a theoretically principled approach pushing straight paths in the latent space of a VAE to approximate geodesic paths along the manifold of the decoded data. Our focus on statistical manifolds—those manifolds whose points correspond to probability distributions of a pre-defined family, defined over the corresponding parameter space $\mathcal{H}$—enables us to draw connections to the theory of VAEs and information geometry. When trained as a likelihood model, a VAE's decoder image spans the manifold's parameter space $\mathcal{H}$. This formulation finds direct application in scRNA-seq, where individual gene counts are assumed to follow a negative binomial distribution, reflecting crucial data properties such as discreteness and overdispersion (Zhou et al., 2011).

Crucially, FlatVI regularises the latent space through a *flattening loss* that pushes the pullback metric from a stochastic VAE decoder towards a spatially-uniform scaled identity matrix, thereby regularising towards latent Euclidean geometry (Fig. 1). In a controlled simulation setting, we demonstrate that our regularisation successfully constrains the latent manifold to exhibit an approximate Euclidean geometry, while enabling likelihood parameter reconstruction on par with standard VAEs. Our method finds direct applications to single-cell representation learning and trajectory inference, which we demonstrate across multiple biological settings by providing an improved data representation for OT-based modelling of latent cellular population dynamics. In summary, we make the following contributions:

- We introduce a regularisation technique for discrete-likelihood VAEs to enforce straight latent interpolations to approximate geodesic paths on the statistical manifold spanned by the decoder.
- We provide an explicit formulation of the flattening loss for the negative binomial case, which directly impacts modelling high-dimensional scRNA-seq data.
- We empirically validate our model on simulated data and latent geodesic interpolations.
- We show that our method offers a better representation space for existing OT-based trajectory inference tools than existing approaches in real data settings.

## 2 RELATED WORK

We discuss related work on geometry-aware representation learning via Autoencoders (AE), as well as OT methods for modelling population dynamics with applications in scRNA-seq.

**Geometry and AEs.** Prior work by Arvanitidis et al. (2020) introduced optimal latent paths reflecting observation space geometry in deterministic and Gaussian stochastic decoders, extended by Arvanitidis et al. (2021) to VAEs with arbitrary likelihoods. Meanwhile, Chen et al. (2020) explored representation learning benefits by modelling latent spaces of deterministic AEs as flat manifolds, while other studies incorporate data geometry via isometric (Yonghyeon et al., 2021) and Jacobian (Nazari et al., 2023) regularisations.

**Geometry in single-cell representations.** While latent variable models for scRNA-seq data are established (Lopez et al., 2018; Eraslan et al., 2019), enforcing geometric structures in single-cell latent spaces is underexplored. Combining single-cell representations and geometry, diffusion-based manifold learning (Moon et al., 2019; Huguet et al., 2024; Fasina et al., 2023) offers insights into geometry-aware low-dimensional representations. Investigating single-cell geometry extends to gene expression data (Korem et al., 2015; Qiu et al., 2022) and dynamic settings (Rifkin & Kim, 2002).

**OT for single-cell RNA-seq.** OT methods in single-cell transcriptomics have shown potential in multiple application contexts, such as trajectory inference (Schiebinger et al., 2019; Lange et al.,

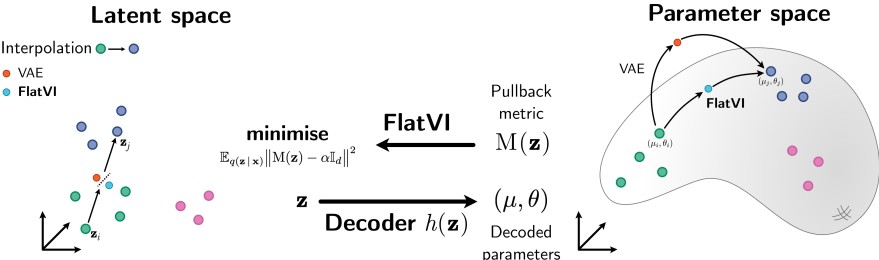

Figure 1: Visual conceptualisation of the FlatVI approach. The decoder of a VAE spans the parameter space of a statistical manifold of probability distributions. In standard VAE settings, straight latent paths are not guaranteed to map to meaningful statistical manifold interpolations through the decoder. By regularising the pullback metric of the stochastic decoder, FlatVI enforces correspondence between straight paths in the latent space and geodesic interpolations along the statistical manifold of the decoded space.

2023), spatial reconstruction (Moriel et al., 2021) and multi-modal alignment (Demetci et al., 2022). Recent efforts have resorted to predicting the effects of drug perturbations on cells (Hetzel et al., 2022) using neural OT (Bunne et al., 2023) and cell trajectories in an unbalanced setting (Eyring et al., 2022). The works from Tong et al. (2020) and Huguet et al. (2022) explore the applicability of OT for modelling gene expression through time. Specifically, Huguet et al. (2022) model continuous single-cell trajectories in a latent space of a Geodesic Autoencoder (GAE) where distances are regularised to approximate the geodesic distances of single-cell data. Recently, Haviv et al. (2024) have introduced a framework for regularising latent distances in AEs to approximate Wasserstein distances between point clouds in the data space, with promising applications in representation learning for spatially-resolved RNA-seq. Finally, our latent interpolations rely on Flow Matching (Lipman et al., 2022), whose OT-based formulation has proved promising at reconstructing trajectories in low-dimensional and continuous cellular representations (Tong et al., 2023; Kapusniak et al., 2024).

## 3 BACKGROUND

### 3.1 VAEs FOR DISCRETE DATA WITH APPLICATIONS IN SCRNA-SEQ

In this work, we deal with discrete count data, formally collected in a high-dimensional matrix $\mathbf{X} \in \mathbb{N}_0^{N \times G}$, where $N$ represents the number of observations and $G$ the number of features. We assume that individual sample features $x_{ng}$ are independent realisations of a discrete random variable $\mathbf{X}_{ng} \sim \mathbb{P}(\cdot | \varphi_{ng})$ with observation-specific parameters $\varphi_{ng}$. Let $\mathbf{x} \in \mathcal{X} = \mathbb{N}_0^G$ be an observation. We consider a joint latent variable model factorising as $p_\phi(\mathbf{x}, \mathbf{z}) = p_\phi(\mathbf{x}|\mathbf{z})p(\mathbf{z})$, where $\mathbf{z} \in \mathcal{Z} = \mathbb{R}^d$ is a $d$-dimensional latent random variable with $d < G$ and $\mathbf{z} \sim p(\mathbf{z})$, with $p(\mathbf{z}) = \mathcal{N}(\mathbf{0}, \mathbb{I}_d)$. Here, $\mathbb{I}_d$ is the squared identity matrix with dimension $d$.

The decoder model is a discrete distribution from a pre-defined family with parameters expressed as a function of the latent variable $\mathbf{z}$ as:

$$p_\phi(\mathbf{x}|\mathbf{z}) = \mathbb{P}(\mathbf{x}|h_\phi(\mathbf{z})) , \tag{1}$$

In deep latent variable models, $h_\phi$ is a deep neural network termed *decoder*. VAEs additionally add an *encoder* network $f_\psi : \mathcal{X} \to \mathcal{Z}$ optimised jointly with $h_\phi$ through the Evidence Lower Bound Objective (ELBO), which introduces the following loss:

$$\mathcal{L}_{\text{ELBO}} = \mathbb{E}_{q_\psi(\mathbf{z}|\mathbf{x})}\big[\log p_\phi(\mathbf{x}|\mathbf{z})\big] - \text{KL}\big(q_\psi(\mathbf{z}|\mathbf{x})\|p(\mathbf{z})\big) , \tag{2}$$

where the first term denotes the likelihood of the data under the decoder model and the second term is the Kullback-Leibler divergence between a Gaussian variational posterior distribution $q_\psi$ parameterised by $f_\psi$ and a standard multivariate Gaussian prior (Kingma, 2013). In other words, as long as one can select a parametric family of distributions $\{\mathbb{P}(\cdot|\varphi)\}_{\varphi \in \mathcal{H}}$ as a reasonable noise model for the dataset properties, the likelihood of the data can be modelled by the decoder of a VAE.

In the field of scRNA-seq, biological and technical variability causes sparsity and overdispersion properties in the expression counts, making the negative binomial likelihood a natural choice for modelling gene expression. **Sparsity** arises from technical limitations in detecting gene transcripts or from unexpressed genes in specific conditions. **Overdispersion** refers to genes having higher variance than the mean, deviating from a Poisson model. This is influenced by technical factors and modelled by the inverse dispersion parameter of a negative binomial distribution (Heumos et al., 2023). Thus, we assume $\mathbb{P}(\varphi_{ng}) = \mathrm{NB}(\mu_{ng}, \theta_g)$, where $\mu_{ng}$ and $\theta_g$ represent the cell-gene-specific mean and the gene-specific inverse dispersion parameters, respectively. In the VAE setting, given a gene-expression vector $\mathbf{x}$, we define the following parameterizations (Lopez et al., 2018):

$$\mathbf{z} = f_\psi(\mathbf{x}), \quad \boldsymbol{\mu} = h_\phi(\mathbf{z}, l) = l\,\mathrm{softmax}(\rho_\phi(\mathbf{z})) , \tag{3}$$

where $\rho_\phi : \mathbb{R}^d \to \mathbb{R}^G$ models expression proportions of individual genes and $l$ is the observed cell-specific size factor directly derived from the data as a cell's total number of counts $l = \sum_{g=1}^{G} x_g$. The encoder $f_\psi$ already takes into account the reparametrisation trick (Kingma, 2013). In what follows, we drop the dependency of $h$ on $\phi$ for notational simplicity.

## 3.2 THE GEOMETRY OF (VARIATIONAL) AUTOENCODERS

**Deterministic Autoencoders.** A common assumption is that continuous data lies near a low-dimensional Riemannian manifold $\mathcal{M}_\mathcal{X}$ associated with a latent representation $\mathcal{Z}$ and with $\mathcal{X} = \mathbb{R}^G$. The decoder $h$ of a deterministic AE model can be seen as an immersion $h : \mathbb{R}^d \to \mathbb{R}^G$ of the latent space $\mathcal{Z} = \mathbb{R}^d$ into the embedded Riemannian manifold $\mathcal{M}_\mathcal{X}$ equipped with a metric tensor M.

**Definition 3.1.** A Riemannian manifold is a smooth manifold $\mathcal{M}_\mathcal{X}$ endowed with a Riemannian metric $\mathrm{M}(\mathbf{x})$ for $\mathbf{x} \in \mathcal{M}_\mathcal{X}$. $\mathrm{M}(\mathbf{x})$ is a positive-definite matrix that changes smoothly and defines a local inner product on the tangent space $\mathcal{T}_\mathbf{x} \mathcal{M}_\mathcal{X}$ as $\langle \mathbf{u}, \mathbf{v} \rangle_{\mathcal{M}_\mathcal{X}} = \mathbf{u}^\mathrm{T} \mathrm{M}(\mathbf{x})\mathbf{v}$, with $\mathbf{v}, \mathbf{u} \in \mathcal{T}_\mathbf{x} \mathcal{M}_\mathcal{X}$ (Do Carmo & Flaherty Francis, 1992).

From Definition 3.1, it derives that a Riemannian manifold in the decoded space has Euclidean geometry when $\mathrm{M}(\mathbf{x}) = \mathbb{I}_G$ everywhere. In this setting, the geometry of the latent space is directly linked to the geometry of observation space by the *pullback metric* $\mathrm{M}(\mathbf{z})$ (Arvanitidis et al., 2021):

$$\mathrm{M}(\mathbf{z}) = \mathbb{J}_h(\mathbf{z})^\mathrm{T} \mathrm{M}(\mathbf{x}) \mathbb{J}_h(\mathbf{z}) , \tag{4}$$

where $\mathbf{x} = h(\mathbf{z})$ and $\mathbb{J}_h(\mathbf{z})$ is the Jacobian matrix of $h(\mathbf{z})$. Here, the decoder $h$ is assumed to be a diffeomorphism between the latent space and its image, such that $\mathbb{J}_h(\mathbf{z})$ is full rank for all $\mathbf{z}$. The existence of a metric $\mathrm{M}(\mathbf{z})$ identifies a latent Riemannian manifold $\mathcal{M}_\mathcal{Z}$, whose properties are defined based on the geometry of $\mathcal{M}_\mathcal{X}$ through Eq. (4). For example, Eq. (4) allows to define the shortest curve $\gamma(t)$ connecting pairs of latent codes $\mathbf{z}_1$ and $\mathbf{z}_2$ as the one minimising the distance between their images $h(\mathbf{z}_1)$ and $h(\mathbf{z}_2)$ on $\mathcal{M}_\mathcal{X}$. More formally:

$$d_{\mathrm{latent}}(\mathbf{z}_1, \mathbf{z}_2) = \inf_{\gamma(t)} \int_0^1 \|\dot{h}(\gamma(t))\| \mathrm{d}t = \inf_{\gamma(t)} \int_0^1 \sqrt{\dot{\gamma}(t)^\mathrm{T} \mathrm{M}(\gamma(t))\dot{\gamma}(t)} \mathrm{d}t , \tag{5}$$

$$\text{where} \quad \gamma(0) = \mathbf{z}_1, \ \gamma(1) = \mathbf{z}_2 .$$

Here, $\gamma(t) : \mathbb{R} \to \mathcal{Z}$ is a curve in the latent space with boundary conditions $\gamma(0) = \mathbf{z}_1$ and $\gamma(1) = \mathbf{z}_2$, and $\dot{\gamma}(t)$ its derivative along the manifold. Crucial to this work, one notices that combining Eq. (5) and Definition 3.1, *when the the metric tensor $\mathrm{M}(\mathbf{z}) = \mathbb{I}_d$, the curve $\gamma^*(t)$ minimising Eq. (5) is the straight line between latent codes*.

**Variational Autoencoders.** While in AEs one deals with deterministic manifolds, in VAEs the decoder function $h$ maps a latent code $\mathbf{z} \in \mathcal{Z}$ to the parameter configuration $\phi \in \mathcal{H}$ of the data likelihood. If the likelihood has continuous parameters, $\mathcal{H} = \mathbb{R}^G$ represents the parameter space. As such, the *decoder image lies on a statistical manifold*, which is a smooth manifold of probability distributions. Such manifolds have a natural metric tensor called *Fisher Information Metric* (FIM) (Nielsen, 2020; Arvanitidis et al., 2021). The FIM defines the local geometry of the statistical manifold and can be used to build the pullback metric for arbitrary decoders. For a statistical manifold $\mathcal{M}_\mathcal{H}$ with parameters $\phi \in \mathcal{H}$, the FIM is formulated as

$$\mathrm{M}(\phi) = \mathbb{E}_{p(\mathbf{x}|\phi)} \left[ \nabla_\phi \log p(\mathbf{x}|\phi) \nabla_\phi \log p(\mathbf{x}|\phi)^\mathrm{T} \right] , \tag{6}$$

where $\phi = h(\mathbf{z})$ and the metric tensor $\mathrm{M}(\phi) \in \mathbb{R}^{G \times G}$. Analogous to deterministic AEs, one can combine Eq. (6) and Eq. (4) to formulate the pullback metric for an arbitrary statistical manifold,

with the difference that the metric tensor is defined based on the parameter space $\mathcal{H}$. Thus, the latent space of a VAE is endowed with the pullback metric for a statistical manifold.

$$\mathrm{M}(\mathbf{z}) = \mathbb{J}_h(\mathbf{z})^\mathrm{T}\mathrm{M}(\boldsymbol{\phi})\mathbb{J}_h(\mathbf{z}) \,, \tag{7}$$

where $\mathrm{M}(\mathbf{z}) \in \mathbb{R}^{d \times d}$. Note that the calculation of the FIM is specific for the chosen likelihood type and, as such, depends on initial assumptions on the data distribution.

### 3.3 LEARNING POPULATION DYNAMICS WITH OPTIMAL TRANSPORT

The complexity of learning trajectories in high-dimensional data can be prevented by interpolating latent representations and decoding intermediate results to the data space for inspection. Here, we deal with learning population dynamics, which consists of modelling the temporal evolution of a dynamical system from unpaired samples of observations through time. As such a task is naturally formulated as a distribution matching problem, dynamic OT has been a popular avenue for population dynamics.

Let the data be defined on a continuous space $\mathcal{X} = \mathbb{R}^d$. OT computes the most efficient mapping for transporting mass from one measure $\nu$, to another $\eta$, defined on $\mathcal{X}$. Relevant to dynamical systems, Benamou & Brenier (2000) introduced a *continuous formulation* of the OT problem. In this setting, let $p_t$ be a time-varying density over $\mathbb{R}^d$ constrained by $p_0 = \nu$ and $p_1 = \eta$. Dynamic OT learns a time-dependent marginal vector field $u : [0, 1] \times \mathbb{R}^d \to \mathbb{R}^d$, where $u_t(\mathbf{x}) = u(t, \mathbf{x})$. Such a field is associated with an ordinary differential equation (ODE) $\mathrm{d}\mathbf{x} = u_t(\mathbf{x})\mathrm{d}t$ whose solution matches the source with the target distribution. Therefore, one can use dynamic OT to learn a system's dynamics from snapshots of data collected over time.

An efficient simulation-free formulation of dynamic OT comes from the OT Conditional Flow Matching (OT-CFM) model by Tong et al. (2023), who demonstrated that the time-resolved marginal vector field $u_t(\mathbf{x})$ has the same minimiser as the data-conditioned vector field $u_t(\mathbf{x}|\mathbf{x}_0, \mathbf{x}_1)$, where $(\mathbf{x}_0, \mathbf{x}_1) \sim q(\mathbf{x}_0, \mathbf{x}_1) = \pi(\mathbf{X}_0, \mathbf{X}_1)$ are tuples of points sampled from the static OT coupling $\pi$ between source and target batches, $\mathbf{X}_0$ and $\mathbf{X}_1$. Assuming Gaussian marginals $p_t$ and $\mathbf{x}_0$ and $\mathbf{x}_1$ to be connected by Gaussian flows, both $p_t(\mathbf{x}|\mathbf{x}_0, \mathbf{x}_1)$ and $u_t(\mathbf{x} \,|\, \mathbf{x}_0, \mathbf{x}_1)$ become tractable:

$$p_t(\mathbf{x} \,|\, \mathbf{x}_0, \mathbf{x}_1) = \mathcal{N}(t\mathbf{x}_1 + (1 - t)\mathbf{x}_0, \sigma^2) \tag{8}$$

$$u_t(\mathbf{x} \,|\, \mathbf{x}_0, \mathbf{x}_1) = \mathbf{x}_1 - \mathbf{x}_0 \,, \tag{9}$$

where the value of $\sigma^2$ is a small pre-defined constant. Accordingly, the OT-CFM loss is

$$\mathcal{L}_{\text{OT-CFM}} = \mathbb{E}_{t, q(\mathbf{x}_0, \mathbf{x}_1), p_t(\mathbf{x}|\mathbf{x}_0, \mathbf{x}_1)} \big[\|v_\xi(t, \mathbf{x}) - u_t(\mathbf{x}|\mathbf{x}_0, \mathbf{x}_1)\|^2\big] \,, \quad \text{with } t \sim \mathcal{U}(0, 1). \tag{10}$$

Given this formulation of dynamic OT, we highlight three aspects: (i) Dynamic OT only applies to continuous spaces. (ii) OT-CFM benefits from low-dimensional representations since the OT-coupling is optimised from distances in the state space. (iii) Based on Eq. (8), OT-CFM uses straight lines to optimise the conditional vector field, thus *assuming Euclidean geometry*.

In the presence of discrete data like scRNA-seq counts, one can tackle (i) and (ii) by learning dynamics in a low-dimensional representation of the state space–the latent space of a VAE with a discrete-likelihood decoder. Note, however, that (iii) is still a shortcoming, since straight lines in the latent space of a VAE do not reflect geodesic paths on the statistical data manifold unless enforced otherwise, see Fig. 1. In this work, we tackle the latter problem through a VAE regularisation approach.

## 4 THE FLATVI MODEL

### 4.1 LEARNING SINGLE-CELL POPULATION DYNAMICS

We are concerned with learning population dynamics along the statistical manifold spanned by the decoder of a VAE—for example, a negative binomial manifold in scRNA-seq—, where observations are collected in $T$ unpaired distributions $\{\nu_t\}_{t=0}^T$. Individual time points correspond to separate snapshot datasets $\{\mathbf{X}_t\}_{t=0}^T$, each with $N_t$ observations. In latent trajectory modelling, each snapshot is mapped to a collection $\{\mathbf{Z}_t\}_{t=0}^T$ of latent representations $\mathbf{Z}_t \in \mathbb{R}^{N_t \times d}$ following the setting described in Sec. 3.1. We wish to use optimal transport with OT-CFM to learn the dynamics of the system through a parameterised vector field $v_\xi(t, \mathbf{z})$ in the latent space $\mathcal{Z}$, such that trajectories in $\mathcal{Z}$

respect the non-Euclidean geometry of the decoded manifold. To achieve such a correspondence, one can either: (i) act on the OT-CFM algorithm to enforce matching distributions via statistical manifold interpolations rather than straight paths or (ii) regularise the representation space, such that linear latent paths correspond to geodesic paths.

**(i)** The ideal approach would be to replace straight interpolations with geodesic curves in the definition of the OT-CFM objective, ensuring that trajectories in the latent space reflect the data geometry. Given a VAE decoder spanning a statistical manifold with metric tensor as in Eq. (7), computing the optimal coupling $\pi$ between $\mathbf{Z}_t$ and $\mathbf{Z}_{t+1}$ to perform OT-CFM requires calculating the geodesic distance in Eq. (5) between all pairs of observations from the consecutive snapshots. However, estimating $\gamma(t)$ for all pairs of observations in consecutive batches is *unfeasible in large datasets* like in scRNA-seq, as it requires solving an optimisation problem to find the optimal $\gamma^*(t)$ connecting all pairs of observations in source and target batches.

**(ii)** Alternatively, one could seek to regularise the latent space of a VAE in such a way that the latent manifold $\mathcal{M}_{\mathcal{Z}}$ has Euclidean geometry, while the non-linear stochastic decoder reintroduces the complex geometry of the statistical manifold $\mathcal{M}_{\mathcal{H}}$ of the decoded space, pushed by the minimisation of the reconstruction loss. In other words, *the goal is to set a correspondence between straight paths in the latent space and geodesic interpolations along the statistical manifold*. If this is verified, decoded trajectories generated using Euclidean OT-CFM in the latent space of the VAE do not violate the geometry of the data manifold. Motivated by applications to single-cell VAEs, we opt for (ii).

## 4.2 FLATTENING LOSS

To ensure that straight latent paths model geodesics along the decoded statistical manifold, we introduce a regularisation to the standard VAE objective which enforces a constant Euclidean local geometry in the latent manifold $\mathcal{M}_{\mathcal{Z}}$. Recall from Sec. 3.2 that the latent space of a VAE has local geometry denoted by the metric in Eq. (7) which is a function of the Fisher information of the decoder's likelihood given the decoded parameters $\phi \in \mathcal{H}$. From Eq. (5) we also know that if $\mathrm{M}(\mathbf{z}) = \mathbb{I}_d$, then the geodesic distance between each pair of latent points is given by the straight line between them. Therefore, regularising the product $\mathbb{J}_h(\mathbf{z})^{\mathrm{T}}\mathrm{M}(\phi)\mathbb{J}_h(\mathbf{z})$ towards $\mathbb{I}_d$ forces a VAE to model Euclidean latent geometry. Crucially, the non-linear decoder is still trained to reconstruct the original data space under the likelihood optimisation task in the ELBO (Eq. (2)), reinstating the local geometry of the decoded statistical manifold described by $\mathrm{M}(\phi)$.

In summary, we implement a *flattening loss* $\mathcal{L}_{\text{flat}}$ to induce latent Euclidean geometry in VAEs with the decoder modelling the data likelihood. Our regularisation is defined as follows:

$$\mathcal{L}_{\text{flat}} = \mathbb{E}_{q_\psi(\mathbf{z}|\mathbf{x})}\left\|\mathrm{M}(\mathbf{z}) - \alpha\mathbb{I}_d\right\|_2^2. \tag{11}$$

Here, $\mathrm{M}(\mathbf{z})$ is calculated by Eq. (7), hence it depends on the Fisher information $\mathrm{M}(\phi)$ of the decoder's likelihood and the Jacobian of the decoder $\mathbb{J}_h(\mathbf{z})$. Meanwhile, $\alpha$ is a trainable parameter offering some flexibility on the scale of the diagonal constraint. In VAEs, the loss of FlatVI is combined with the ELBO:

$$\mathcal{L}_{\text{FlatVI}} = \mathcal{L}_{\text{ELBO}} + \lambda\mathcal{L}_{\text{flat}}, \tag{12}$$

where $\lambda$ controls the strength of the flattening regularisation. On synthetic data, one can show that such an approach yields better manifold interpolations (more in App. I.2). In our real-world single-cell experiments on learning population dynamics, we first train a VAE with regularisation as in Eq. (12), then we use the resulting Euclidean representation space for latent OT-CFM. The procedure used to train FlatVI is summarised in Algorithm 1, whereas its combination with OT-CFM is illustrated in Algorithm 2.

**Remarks.** Both the reconstruction loss in the ELBO and the flattening objective influence the decoder parameters. This is because both the Fisher information and Jacobian depend on the decoded likelihood parameters. Thus, training FlatVI involves updating the decoder to both reconstruct the data and produce a scaled identity as a latent metric. Additionally, note that evaluating the flattening loss is slower than the usual VAE loss, as it requires differentiating through the decoder to compute the FIM.

### 4.3 APPLICATION: FLATVI ON A NEGATIVE BINOMIAL SINGLE-CELL MANIFOLD

In this work, we are interested in modelling cellular trajectories to study the evolution of biological processes. As outlined in Sec. 3.1, single-cell counts are modelled with a negative binomial decoder with the following univariate point mass function for each gene $g$ independently:

$$p_{\text{NB}}(x_g|\mu_g, \theta_g) = C \left(\frac{\theta_g}{\theta_g + \mu_g}\right)^{\theta_g} \left(\frac{\mu_g}{\theta_g + \mu_g}\right)^{x_g}, \text{where } C = \frac{\Gamma(\theta_g + x_g)}{x_g!\Gamma(\theta_g)}, \ \mu, \theta > 0$$

with $x_g \in \mathbb{N}_0$ and $\mu_g = h_g(\mathbf{z})$. Notably, since the decoder $h$ produces cell-specific means, each cell is deemed as an individual probability distribution. Consequently, we assume that the data lies on a statistical manifold parameterised by the decoder in the space of negative binomial distributions. According to Eq. (7), we pull back the FIM of the statistical manifold of the negative binomial probability distribution to the latent manifold $\mathcal{M}_{\mathcal{Z}}$.

**Proposition 4.1.** *The pullback metric evaluated at the latent point $\mathbf{z} \in \mathcal{Z}$ of a negative binomial statistical manifold parameterised by the mean decoder $h$ and inverse dispersion $\boldsymbol{\theta}$ is*

$$\text{M}(\mathbf{z}) = \sum_g \frac{\theta_g}{h_g(\mathbf{z})(h_g(\mathbf{z}) + \theta_g)} \nabla_{\mathbf{z}} h_g(\mathbf{z}) \otimes \nabla_{\mathbf{z}} h_g(\mathbf{z}) , \tag{13}$$

*where $\otimes$ is the outer product of vectors and $g$ indexes individual decoded dimensions.*

We provide the proof of Proposition 4.1 in App. A. Note that we only take the gradient of the mean decoder $h$ since the inverse dispersion parameter is not a function of the latent space. We use the definition of $\text{M}(\mathbf{z})$ in Eq. (13) to apply the flattening loss in Eq. (11) to single-cell data and learn latent cellular trajectories downstream using Euclidean OT-CFM. In App. D, we also include an estimate of the complexity for computing the FIM.

## 5 EXPERIMENTS

We show that FlatVI improves both real and simulated discrete data. Our regularization enhances the approximation of constant Euclidean geometry in the latent manifold while preserving accurate likelihood parameter reconstruction on synthetic data. FlatVI also boosts single-cell data representation and trajectory inference, especially when paired with Euclidean OT. Lastly, we demonstrate its effectiveness as a model for learning biologically relevant vector fields on the cellular manifold.

### 5.1 SIMULATED DATA

**Task and datasets.** To demonstrate the properties of our model, we evaluate the effect of FlatVI on latent representations using a multivariate negative binomial synthetic dataset. Our goal is to establish the successful induction of Euclidean geometry in the latent space of the VAE while preserving data reconstruction. We simulate 1000 observations from a 10-dimensional negative binomial distribution with known mean ($\boldsymbol{\mu}$) and inverse dispersion ($\boldsymbol{\theta}$). To emulate a real scenario, we generate cells from three distinct categories, representing biological cell types (see Fig. 5 for the PCA representation of the data). A cell-type label is

Table 1: Comparison between FlatVI and the unregularised NB-VAE ($\lambda = 0$) in terms of likelihood parameter reconstruction (MSE ($\boldsymbol{\mu}$) and ($\boldsymbol{\theta}$)) and geodesic path flattening. The latter metric is the MSE between latent Euclidean and geodesic distances computed using Eq. (5) between 1000 couples of simulated observations.

| Reg. strength $\lambda$ | MSE ($\boldsymbol{\mu}$) ($\downarrow$) | MSE ($\boldsymbol{\theta}$) ($\downarrow$) | MSE (Geo-Euc) ($\downarrow$) |
|---|---|---|---|
| $\lambda = 0$ | $15.52_{\pm 0.94}$ | $3.10_{\pm 0.19}$ | $47.75_{\pm 2.80}$ |
| $\lambda = 1$ | $16.34_{\pm 0.46}$ | $5.67_{\pm 0.88}$ | $46.34_{\pm 6.45}$ |
| $\lambda = 3$ | $16.35_{\pm 0.53}$ | $3.09_{\pm 0.31}$ | $16.74_{\pm 2.32}$ |
| $\lambda = 5$ | $\mathbf{14.75_{\pm 0.12}}$ | $3.20_{\pm 0.20}$ | $8.65_{\pm 1.68}$ |
| $\lambda = 7$ | $15.47_{\pm 0.20}$ | $3.38_{\pm 0.09}$ | $\mathbf{8.02_{\pm 0.67}}$ |
| $\lambda = 10$ | $15.41_{\pm 0.07}$ | $\mathbf{3.08_{\pm 0.13}}$ | $11.80_{\pm 1.04}$ |

drawn uniformly from the three available classes, each defined by a different mean distribution. The means of negative binomial observations across categories are sampled from normal distributions with centres at -1, 0, and 1, each with a standard deviation of 1. These means are exponentiated to ensure positivity. Gene-specific inverse dispersion parameters are sampled from a Gamma distribution (concentration 2, rate 1) and made positive via absolute value. All cells share the same gene-specific dispersion parameters. More information about the simulation setting is in App. F.2.

**Evaluation.** For regularisation strengths $\lambda \in \{0, 1, 3, 5, 7, 10\}$ we evaluate: (i) The Mean Squared Error (MSE) reconstruction of the mean $\boldsymbol{\mu}$ and inverse dispersion $\boldsymbol{\theta}$ of the simulated data obtained by

training the regularised VAE and decoding the latent representations. (ii) The MSE between geodesic and Euclidean distances between couples of latent codes. The geodesics between pairs of latent codes are learnt as cubic splines minimising the curve length illustrated in Eq. (5) using Eq. (7) as a metric tensor. A successful training recovers the true parameters of the data-generating process while imposing proximity between the Euclidean and pullback-based geodesic distances between latent codes. Furthermore, we visually evaluate two Riemannian metrics in the absence of regularisation and when $\lambda = 7$ [1]: The *Variance of the Riemannian metric* (VoR) and the *condition number* (CN) following Chen et al. (2020) and Yonghyeon et al. (2021). Briefly, given a pullback metric tensor $\mathrm{M}(\mathbf{z})$, the VoR quantifies the uniformity of the Riemannian metric across space by computing the distance between $\mathrm{M}(\mathbf{z})$ and $\bar{\mathrm{M}} = \mathbb{E}_{\mathbf{z} \sim p_{\mathbf{z}}}[\mathrm{M}(\mathbf{z})]$. A VoR of 0 indicates constant metric across $\mathcal{Z}$. The CN is the ratio between the maximum and minimum eigenvalues of the Riemannian metric. It is close to 1 when the metric tensor is close to the identity metric. A successful flattening of the latent space involves low VoR and CN close to 1. We provide a more thorough definition of the metrics in App. F.1.

**Results.** In our simulation setting, results in Tab. 1 show that our regularisation forces geodesic distances to better approximate Euclidean distances in the latent space compared to an unregularized NB-VAE, with increasing $\lambda$ reducing the MSE between pairwise geodesic and Euclidean distances. Meanwhile, the capabilities of our model to reconstruct the mean ($\boldsymbol{\mu}$) and inverse dispersion parameter ($\boldsymbol{\sigma}$) do not degrade when the regularisation strength is increased. The plots in Fig. 2 serve as additional proof of the flattening mechanism. Inducing Euclidean geometry into the latent space ensures a more uniform local geometry, as the latent manifold of FlatVI does not exhibit as many regions of systematically high VoR or CN as in the standard NB-VAE setting (see Fig. 2a-b). Despite the flattening, some limited regions with high CN and VoR remain in the FlatVI embedding. We compare our results with an Euclidean space in App. I.1.2 and investigate the cause for high VoR and CN values in App. I.1.4. In Fig. 2c we sample 10 couples of points from regions of high VoR in the NB-VAE latent space and plot geodesic paths according to Eq. (5) on both FlatVI and its unregularised counterpart. FlatVI achieves straight paths, while pullback-based geodesic interpolations in the standard NB-VAE bottleneck show a curvature.

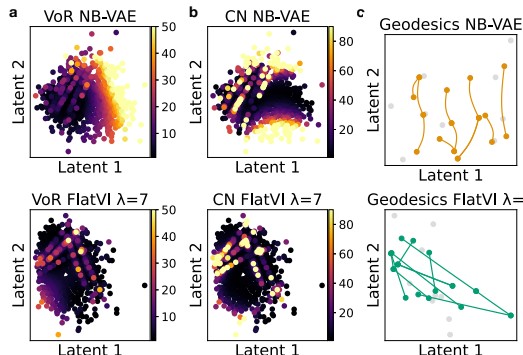

Figure 2: Comparison between the latent geometries of the NB-VAE (top row) and FlatVI trained with $\lambda = 7$ (bottom row) evaluated in terms of **(a)** Variance of the Riemannian metric (VoR), **(b)** Condition Number (CN) and **(c)** Straightness of the geodesic paths connecting pairs of latent points.

### 5.2 RECONSTRUCTION OF SCRNA-SEQ TRAJECTORIES

**Task and dataset.** Our basic hypothesis is that FlatVI's latent space offers a better representation for Euclidean OT than the standard unregularised VAE, as our flattening loss changes the parameters of the decoder to approximate local Euclidean geometry in the latent manifold. Here, we show the advantages of using FlatVI in combination with Euclidean OT-CFM for mapping single-cell trajectories through time on two real datasets: (i) The Embryoid body (EB) (Moon et al., 2019) dataset profiles 18,203 differentiating human embryoid cells over five time points, generating four lineages. (ii) The reprogramming dataset (MEF) (Schiebinger et al., 2019) explores the reprogramming of mouse embryonic fibroblasts into induced pluripotent stem cells, comprising 165,892 cells across 39 time points.

**Baselines.** We compare FlatVI with a standard VAE (NB-VAE) trained with a negative binomial decoder (Lopez et al., 2018) as representation models for continuous OT. Additionally, we evaluate latent OT on the embeddings produced by the GAE model described in Huguet et al. (2022) and introduced in Sec. 2. The latter model is trained on $\log$-normalised gene expression to better accommodate the lack of a discrete probabilistic decoder. More details on the difference between FlatVI and GAE are in App. C.2. All three approaches are used to derive embeddings of time-resolved gene expression datasets. Subsequently, we use the cell representations to train an OT-CFM model and learn latent trajectories using unpaired batches of observations from consecutive time points.

---

[1]Chosen based on the flattening results in Tab. 1.

Table 2: Comparison of cellular trajectory reconstruction on held-out time points using different representation models. Latent trajectories are learnt with OT-CFM leaving out intermediate time points and using them as ground truth for evaluating the interpolation of the cellular dynamics. Distribution matching metrics are evaluated to compare real held-out points and reconstructions thereof in both the decoded and latent space across three seeds.

| | EB | | | | | | MEF | | | | | |
| | Latent | | Decoded | | | | Latent | | Decoded | | | |
| | WD ($\downarrow$) | L2 ($\downarrow$) | D ($\uparrow$) | C ($\uparrow$) | MMD ($\downarrow$) | WD ($\downarrow$) | WD ($\downarrow$) | L2 ($\downarrow$) | D ($\uparrow$) | C ($\uparrow$) | MMD ($\downarrow$) | WD ($\downarrow$) |
|---|---|---|---|---|---|---|---|---|---|---|---|---|
| GAE | 2.16$_{\pm0.14}$ | 0.40$_{\pm0.06}$ | 0.05$_{\pm0.00}$ | 0.02$_{\pm0.00}$ | 0.14$_{\pm0.04}$ | 70.29$_{\pm0.00}$ | 2.49$_{\pm0.22}$ | 0.57$_{\pm0.07}$ | 0.00$_{\pm0.00}$ | 0.00$_{\pm0.00}$ | 0.38$_{\pm0.01}$ | 106.83$_{\pm0.01}$ |
| NB-VAE | 2.07$_{\pm0.07}$ | 0.30$_{\pm0.02}$ | 0.22$_{\pm0.03}$ | 0.38$_{\pm0.03}$ | 0.09$_{\pm0.01}$ | 43.36$_{\pm0.19}$ | 2.07$_{\pm0.12}$ | 0.40$_{\pm0.05}$ | 0.13$_{\pm0.02}$ | 0.10$_{\pm0.01}$ | 0.19$_{\pm0.01}$ | 103.29$_{\pm0.01}$ |
| FlatVI | **1.54$_{\pm0.09}$** | **0.27$_{\pm0.03}$** | **0.31$_{\pm0.05}$** | **0.49$_{\pm0.02}$** | **0.07$_{\pm0.01}$** | **41.99$_{\pm0.04}$** | **1.64$_{\pm0.13}$** | **0.36$_{\pm0.05}$** | **0.16$_{\pm0.04}$** | **0.13$_{\pm0.01}$** | **0.16$_{\pm0.01}$** | **97.12$_{\pm0.01}$** |

**Evaluation.** In our quantitative evaluation, we assess the quality of the transport map in the latent space of different representation models using the 2-Wasserstein and mean $L^2$ distances between real and reconstructed latent cells per time point. Additionally, we evaluate the mixing of real and predicted cells in the decoded gene expression space using the nearest-neighbour-based *Density and Coverage* ($D$ & $C$) metrics (Naeem et al., 2020), linear-kernel Mean Maximum Discrepancy (MMD) Borgwardt et al. (2006) and 2-Wasserstein distance. Notably, for our evaluation, we adopt a similar strategy as Tong et al. (2020). For each dataset, we leave out intermediate time points and train OT-CFM on the remaining cells. The capacity of OT to reconstruct unseen time point $t$ from $t-1$ during inference is an indication of the interpolation abilities of the model along the data manifold. Here, we use such an evaluation paradigm to compare different representation spaces. A value of $\lambda = 1$ is chosen for the EB dataset, while for the MEF reprogramming setting, we set $\lambda = 0.1$. The hyperparameter is tuned based on the value that leads to the best representation for trajectory reconstruction on training data (see App. E for more details).

**Results.** In Tab. 2, we report the reconstruction metrics between true and interpolated latent cells. On all datasets and metrics, trajectories in FlatVI's Euclidean latent space yield better overall latent time point reconstruction results compared to the baseline representation models, valuing the contribution of our regularisation choices. Furthermore, the experiments show that our approach yields an overall improvement in the inferred decoded trajectories in the count space, which can be seen by higher Density and Coverage metrics and lower MMD and 2-Wasserstein distances across all evaluated datasets.

## 5.3 LATENT VECTOR FIELD AND LINEAGE MAPPING

**Task and dataset.** We evaluate the capacity of continuous OT to identify a biologically meaningful cell velocity field using the representation spaces computed by FlatVI, the unregularised NB-VAE and the GAE model. We hypothesise that dynamic OT with Euclidean cost benefits from being applied to a flat representation space. As a dataset for the analysis, we employ the Pancreatic endocrinogenesis (Pancreas) by Bastidas-Ponce et al. (2019), which measures 16,206 cells and spans embryonic days 14.5 to 15.5, revealing multipotent cell differentiation into endocrine and non-endocrine lineages. More specifically, we train the compared representation learning frameworks on the dataset and learn separate vector fields for all models' embeddings matching days 14.5 to 15.5 with OT-CFM. The learnt vector field represents the directionality of the observations on the cellular development manifold.

**Evaluation.** Using the CellRank model (Lange et al., 2022; Weiler et al., 2023), we build random walks on a cell graph based on the directionality of latent velocities learnt by OT-CFM in the different representation spaces. Walks converge to macrostates representing the endpoints of the biological process if the learnt velocity field points to biologically meaningful directions. We quantify the quality of vector fields learnt by OT in different latent spaces based on (i) the number of macrostates identified by random walks, (ii) the velocity consistency, measured as the correlation of the latent velocity field of single datapoints with that of the neighbouring cells. Higher consistency indicates smoother transitions in the vector field, suggesting that the representation space facilitates more coherent and biologically meaningful dynamics, making it a suitable space for learning trajectories (see App. F.1).

**Results.** Figure 3a summarises the number of terminal cell states identified by following the velocity graph. From prior biological knowledge, it is known that the dataset contains six terminal states, which are all identified on the representation computed by our FlatVI ($\lambda = 1$). In contrast, on the GAE and NB-VAE's representations, CellRank only captures four and five terminal states, respectively. In Figure 3b, we further evaluate the velocity consistency within neighbourhoods of cells as a function of latent dimensionality. In line with previous results, OT on the Euclidean latent space yields a more consistent velocity field across latent dimensionalities.

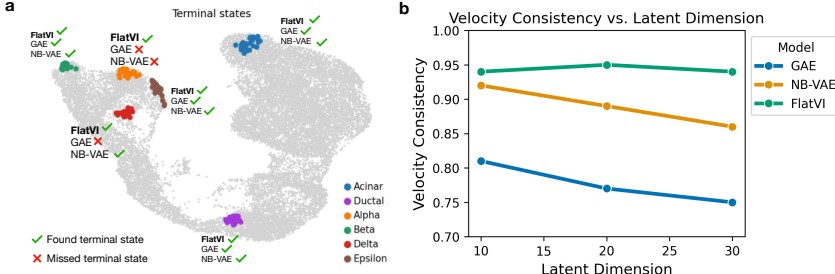

Figure 3: Learning terminal states from OT-CFM's cell velocities in the Pancreas dataset. (**a**) Comparison of terminal states found by CellRank using 10-dimensional latent spaces. (**b**) Consistency computed for the latent velocities of cells across different latent space dimensions.

### 5.3.1 SINGLE-CELL DATA REPRESENTATIONS

Finally, we visualise single-cell latent representations on the previously introduced datasets computed using the FlatVI, NB-VAE, and GAE models. For FlatVI, the value of $\lambda$ was set to 1 for EB and Pancreas and 0.1 for the MEF dataset, in line with previous settings. In Figure 4, we compare the PCA embeddings of FlatVI's latent space with competing models, highlighting initial and terminal cellular states. Despite the regularisation, FlatVI effectively represents the biological structure in the latent space as illustrated by the separation between initial and terminal states. This is particularly evident in the MEF dataset, where FlatVI provides clearer separation between initial and terminal states, suggesting improved identification of cellular dynamics. In Tab. 7 we show that such a separation is more pronounced than competing models also on the

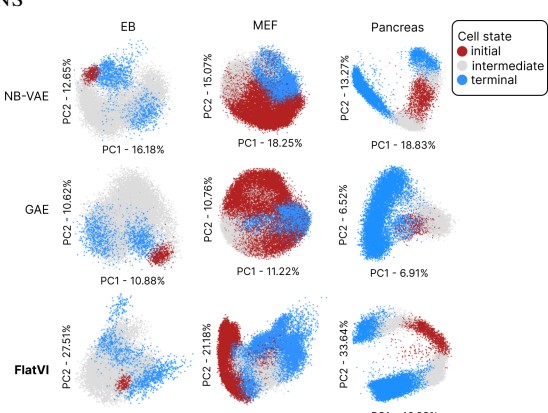

Figure 4: 2D PCA plots of the latent spaces computed by GAE, NB-VAE and FlatVI. Marked are initial, intermediate and terminal cell states along the biological trajectory.

Pancreatic dataset based on quantitative clustering metrics. Moreover, the higher variance explained by individual PCs in FlatVI's latent space suggests that our model captures the main sources of variation (the biological trajectories) more efficiently, reducing latent space dimensionality and enhancing information compression while preserving or even improving biological fidelity. Consequently, FlatVI is well-suited for smaller latent spaces, making it a promising input for OT-based methods.

## 6 CONCLUSION

We addressed modelling temporal trajectories from unpaired cell distributions by using walks on flat NB manifolds. To achieve this, we introduced FlatVI, a VAE training strategy where the pullback metric of the stochastic decoder is regularised to approximate the identity matrix and regularise towards local Euclidean geometry in the latent space. Consequently, straight latent paths correspond to geodesic interpolations in the decoded space. Results on real and synthetic data demonstrate that our flattening procedure holds and enhances the biological structure in the latent space. By combining this approach with dynamic OT, we observed better prediction outcomes and more consistent vector fields on cellular manifolds. These improvements benefit key tasks in cellular development, such as fate mapping and trajectory analysis.

**Limitations and future work.** A limitation of applying FlatVI to real data is that excessive regularisation may cause a trade-off between flattening and reconstruction likelihood. We aim to enhance the model's robustness to reconstruction loss and extend it to a wider range of statistical manifolds and single-cell tasks, such as modelling Poisson-distributed chromatin accessibility, batch correction evaluation, and OT-mediated perturbation modelling.

## 7 ETHICS STATEMENT

The presented work deals with fundamental characteristics of scRNA-seq data and studies how efficient representations of complex high-dimensional cellular data can help to address key biological questions. We envision the release of FlatVI as a user-friendly, open-source tool to enable its widespread use as an option for single-cell analysis. Dealing with biological data, FlatVI could be used in sensitive settings involving clinical information and patient data.

## 8 REPRODUCIBILITY STATEMENT

The details for the reproduction of our work are contained in the Appendix and the main text. The proof for the derivation of the pullback metric of a negative binomial VAE is provided in App. A. A procedural description of the FlatVI model training and its combination with OT-CFM can be found in App. H, more specifically in Algorithm 1 and Algorithm 2. Baselines are detailed in App. B.1. The range of explored hyperparameters is provided in Tab. 3 and details on the model training choices are reported in App. E. All metrics are explained in App. F.1 and experiments are described in detail in App. F.2. All datasets are public and reported with the relative reference publications in App. G.1 and in the main. Pre-processing is outlined in App. G.2. Computational resources are listed in App. G.3.

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

## A    DERIVATION OF THE FISHER INFORMATION METRIC FOR THE NEGATIVE BINOMIAL DISTRIBUTION

We first show that the Fisher Information of a univariate Negative Binomial (NB) distribution parameterised by mean $\mu$ and inverse dispersion $\theta$ with respect to $\mu$ is

$$\mathrm{M}(\mu) = \frac{\theta}{\mu(\mu + \theta)} \; . \tag{14}$$

We then move on with the derivation of the pullback metric in Proposition 4.1.

**Fisher information of the NB distribution.**    The univariate NB probability distribution parameterised by mean $\mu$ and inverse dispersion $\theta$ is

$$p_{\mathrm{NB}}(x \mid \mu, \theta) = \frac{\Gamma(\theta + x)}{x! \Gamma(\theta)} \Big( \frac{\theta}{\theta + \mu} \Big)^{\theta} \Big( \frac{\mu}{\theta + \mu} \Big)^{x} \; . \tag{15}$$

The Fisher information of the distribution can be computed with respect to $\mu$ as:

$$\mathrm{M}(\mu) = -\mathbb{E}_{p(x|\mu,\theta)} \left[ \frac{\partial^2}{\partial \mu^2} \log p_{\mathrm{NB}}(x \mid \mu, \theta) \right] \; . \tag{16}$$

where

$$\log p_{\mathrm{NB}}(x \mid \mu, \theta) = C + \theta \left[ \log(\theta) - \log(\theta + \mu) \right] + x \left[ \log(\mu) - \log(\theta + \mu) \right] \; , \tag{17}$$

with $C = \log(\Gamma(\theta + x)) - \log(x!) - \log(\Gamma(\theta))$. Then, it can be shown that

$$\frac{\partial^2}{\partial \mu^2} \log p_{\mathrm{NB}}(x \mid \mu, \theta) = \frac{\theta + x}{(\theta + \mu)^2} - \frac{x}{\mu^2} \; . \tag{18}$$

Using the fact that the parameterisation involving the mean $\mu$ and inverse dispersion $\theta$ implies that

$$\mathbb{E}_{p(x|\mu,\theta)} [x] = \mu \; , \tag{19}$$

we can expand Eq. (16) as follows

$$
\begin{aligned}
\mathrm{M}(\mu) &= -\mathbb{E}_{p(x|\mu,\theta)} \left[ \frac{\theta + x}{(\theta + \mu)^2} - \frac{x}{\mu^2} \right] \\
&= -\frac{1}{(\theta + \mu)^2} \mathbb{E}_{p(x|\mu,\theta)} [\theta + x] + \frac{1}{\mu^2} \mathbb{E}_{p(x|\mu,\theta)} [x] \\
&= \frac{\theta}{\mu(\mu + \theta)} \; .
\end{aligned}
\tag{20}
$$

**Derivation of the Fisher information metric.**    We here consider the NB-VAE case, where the likelihood is parameterised by $\mu_g = h_g(\mathbf{z})$ and $\theta_g$ independently for each gene $g$.

When $h$ is a continuously differentiable function of $\mathbf{z}$, the pullback metric $\mathrm{M}_g(\mathbf{z})$ of the output $g$ w.r.t $\mathbf{z}$ by the reparameterisation property (Lehmann & Casella, 2006) is

$$
\begin{aligned}
\mathrm{M}_g(\mathbf{z}) &= \nabla_{\mathbf{z}} h_g(\mathbf{z}) \mathrm{M}(h_g(\mathbf{z})) \nabla_{\mathbf{z}} h_g(\mathbf{z})^T \\
&= \frac{\theta_g}{h_g(\mathbf{z})(h_g(\mathbf{z}) + \theta_g)} \nabla_{\mathbf{z}} h_g(\mathbf{z}) \otimes \nabla_{\mathbf{z}} h_g(\mathbf{z}) \; ,
\end{aligned}
\tag{21}
$$

where $\otimes$ is the outer product of vectors, and the gradients are column vectors.

By the chain rule, the joint Fisher information of independent random variables equals the sum of the Fisher information values of each variable (Zamir, 1998). As all $x_g$ are independent given $\mathbf{z}$ in the NB-VAE, the resulting Fisher Information Metric (FIM) is

$$
\begin{aligned}
\mathrm{M}(\mathbf{z}) &= \sum_g \mathrm{M}_g(\mathbf{z}) \\
&= \sum_g \frac{\theta_g}{h_g(\mathbf{z})(h_g(\mathbf{z}) + \theta_g)} \nabla_{\mathbf{z}} h_g(\mathbf{z}) \otimes \nabla_{\mathbf{z}} h_g(\mathbf{z}) \; .
\end{aligned}
\tag{22}
$$

## B  THE GEOMETRY OF AEs

We deal with the assumption that the observed data lies near a Riemannian manifold $\mathcal{M}_\mathcal{X}$ embedded in the ambient space $\mathcal{X} = \mathbb{R}^G$. The manifold $\mathcal{M}_\mathcal{X}$ is defined as follows:

**Definition B.1.** A Riemannian manifold is a smooth manifold $\mathcal{M}_\mathcal{X}$ endowed with a Riemannian metric $\mathrm{M}(\mathbf{x})$ for $\mathbf{x} \in \mathcal{M}_\mathcal{X}$. $\mathrm{M}(\mathbf{x})$ changes smoothly and identifies an inner product on the tangent space $\mathcal{T}_\mathbf{x}\mathcal{M}_\mathcal{X}$ at a point $\mathbf{x} \in \mathcal{M}_\mathcal{X}$ as $\langle \mathbf{u}, \mathbf{v} \rangle_{\mathcal{M}_\mathcal{X}} = \mathbf{u}^\mathbf{T}\mathrm{M}(\mathbf{x})\mathbf{v}$, with $\mathbf{v}, \mathbf{u} \in \mathcal{T}_\mathbf{x}\mathcal{M}_\mathcal{X}$.

For an embedded manifold $\mathcal{M}_\mathcal{X}$ with intrinsic dimension $d$, we can assume the existence of an invertible global chart map $\xi : \mathcal{M}_\mathcal{X} \to \mathbb{R}^d$ mapping the manifold $\mathcal{M}_\mathcal{X}$ to its intrinsic coordinates. A vector $\mathbf{v}_\mathbf{x} \in \mathcal{T}_\mathbf{x}\mathcal{M}_\mathcal{X}$ on the tangent space of $\mathcal{M}_\mathcal{X}$ can be expressed as a pushforward $\mathbf{v}_\mathbf{x} = \mathbb{J}_{\xi^{-1}}(\mathbf{z})\mathbf{v}_\mathbf{z}$ of a tangent vector $\mathbf{v}_\mathbf{z} \in \mathbb{R}^d$ at $\mathbf{z} = \xi(\mathbf{x})$, where $\mathbb{J}$ indicates the Jacobian. Therefore, $\mathbb{J}_{\xi^{-1}}$ maps vectors $\mathbf{v} \in \mathbb{R}^d$ into the tangent space of the embedded manifold $\mathcal{M}_\mathcal{X}$. Following Arvanitidis et al. (2020), the ambient metric $\mathrm{M}(\mathbf{x})$ can be related to the metric $\mathrm{M}(\mathbf{z})$ defined in terms of intrinsic coordinates via:

$$\mathrm{M}(\mathbf{z}) = \mathbb{J}_{\xi^{-1}}(\mathbf{z})^T \mathrm{M}(\xi^{-1}(\mathbf{z}))\mathbb{J}_{\xi^{-1}}(\mathbf{z}) \ . \tag{23}$$

In other words, we can use the metric $\mathrm{M}(\mathbf{z})$ to compute quantities on the manifold, such as geodesic paths. However, for an embedded manifold $\mathcal{M}_\mathcal{X}$, the chart map $\xi$ is usually not known. A workaround is to define the geometry of $\mathcal{M}_\mathcal{X}$ on another Riemannian manifold $\mathcal{M}_\mathcal{Z}$ with a trivial chart map $\xi(\mathbf{z}) = \mathbf{z}$ for $\mathbf{z} \in \mathcal{M}_\mathcal{Z}$, which can be mapped to $\mathcal{M}_\mathcal{X}$ via a smooth immersion $h$. In the next section, we elaborate on the connection between manifold learning and autoencoders following Arvanitidis et al. (2020; 2021).

### B.1  DETERMINISTIC AEs

We assume the decoder $h : \mathcal{Z} = \mathbb{R}^d \to \mathcal{X} = \mathbb{R}^G$ of a deterministic autoencoder is an immersion of the latent space into a Riemannian manifold $\mathcal{M}_\mathcal{X}$ embedded in $\mathcal{X}$ and with metric $\mathrm{M}$. This is valid if one also assumes that $d$ is the intrinsic dimension of $\mathcal{M}_\mathcal{X}$. As explained before, the Jacobian of the decoder maps tangent vectors $\mathbf{v}_\mathbf{z} \in \mathcal{T}_\mathbf{z}\mathcal{M}_\mathcal{Z}$ to tangent vectors $\mathbf{v}_{\mathbf{x}=h(\mathbf{z})} \in \mathcal{T}_\mathbf{x}\mathcal{M}_\mathcal{X}$. The decoder induces a metric into the latent space following Eq. (23) as

$$\mathrm{M}(\mathbf{z}) = \mathbb{J}_h(\mathbf{z})^T \mathrm{M}(h(\mathbf{z}))\mathbb{J}_h(\mathbf{z}) \ , \tag{24}$$

called *pullback metric*. The pullback metric defines the geometry of the latent manifold $\mathcal{M}_\mathcal{Z}$ compared to that of the manifold $\mathcal{M}_\mathcal{X}$. The metric tensor $\mathrm{M}(\mathbf{z})$ regulates the inner product of vectors $\mathbf{u}$ and $\mathbf{v}$ on the tangent space $\mathcal{T}_\mathbf{z}\mathcal{M}_\mathcal{Z}$:

$$\langle \mathbf{u}, \mathbf{v} \rangle_{\mathcal{M}_\mathcal{Z}} = \mathbf{u}^T \mathrm{M}(\mathbf{z})\mathbf{v} \ . \tag{25}$$

To enhance latent representation learning, distances in the latent space $\mathcal{Z}$ can be optimised according to quantities of interest in the observation space $\mathcal{X}$, following the geometry of $\mathcal{M}_\mathcal{X}$. For instance, we can define the length of a curve $\gamma : [0, 1] \to \mathcal{Z}$ in the latent space by measuring its length on the manifold $\mathcal{M}_\mathcal{X}$:

$$L(\gamma) = \int_0^1 \left\| \dot{h}(\gamma(t)) \right\| \mathrm{d}t$$

$$= \int_0^1 \sqrt{\dot{\gamma}(t)^T \mathrm{M}(\gamma(t))\dot{\gamma}(t)}\mathrm{d}t \ , \tag{26}$$

where the equality is derived by applying the chain rule of differentiation.

### B.2  PULLING BACK THE INFORMATION GEOMETRY

In machine learning, exploring latent spaces is crucial, especially in generative models like VAEs. One challenge is defining meaningful distances in the latent space $\mathcal{Z}$, which often depends on the properties of stochastic decoders and their alignment with the observation space. Injecting the geometry of the decoded space of a VAE into the latent space requires a different theoretical framework, where the data is assumed to lie near a statistical manifold.

VAEs can model different kinds of data types by using the decoder function as a non-linear likelihood parameter estimation model. We consider the decoder's output space as a parameter space $\mathcal{H}$ for a probability density function. Depending on the data type, we express a likelihood function $p(\mathbf{x} \mid \boldsymbol{\phi})$ with parameters $\boldsymbol{\phi} \in \mathcal{H}$, reformulated as $p(\mathbf{x} \mid \mathbf{z})$ through a mapping $h : \mathcal{Z} \to \mathcal{H}$. We aim to define a natural distance measure in $\mathcal{Z}$ for infinitesimally close points $\mathbf{z}_1$ and $\mathbf{z}_2 = \mathbf{z}_1 + \delta \mathbf{z}$ when seen from $\mathcal{H}$. Arvanitidis et al. (2021) justify that such a distance corresponds to the Kullback-Leibler (KL) divergence:

$$\text{dist}^2(\mathbf{z}_1, \mathbf{z}_2) = \text{KL}(p(\mathbf{x} \mid \mathbf{z}_1), p(\mathbf{x} \mid \mathbf{z}_2)) . \tag{27}$$

To define the geometry of the statistical manifold, one can resort to information geometry, which studies probabilistic densities represented by parameters $\boldsymbol{\phi} \in \mathcal{H}$. In this framework, $\mathcal{H}$ becomes a statistical manifold equipped with a FIM:

$$\text{M}(\boldsymbol{\phi}) = \int_{\mathcal{X}} [\nabla_{\boldsymbol{\phi}} \log p(\mathbf{x} \mid \boldsymbol{\phi})][\nabla_{\boldsymbol{\phi}} \log p(\mathbf{x} \mid \boldsymbol{\phi})]^T p(\mathbf{x} \mid \boldsymbol{\phi}) \, \mathrm{d}\mathbf{x} . \tag{28}$$

The FIM locally approximates the KL divergence. For a univariate density $p$, parameterised by $\xi$, it is known that

$$\text{KL}(p(x \mid \phi), p(x \mid \phi + \delta \phi)) \approx \frac{1}{2} \delta \phi^{\top} \text{M}(\phi) \delta \phi + o(\delta \phi^2) . \tag{29}$$

In the VAE setting, we view the decoder not as a mapping to the observation space $\mathcal{X}$ but as a transformation to the parameter space $\mathcal{H}$. This perspective allows us to naturally incorporate the FIM into the latent space $\mathcal{Z}$. Consequently, the VAE's decoder can be seen as spanning a manifold $\mathcal{M}_{\mathcal{H}}$ in $\mathcal{H}$, with $\mathcal{M}_{\mathcal{Z}}$ inheriting the metric in Eq. (28) via the Riemannian pullback. Based on this, we define a statistical manifold.

**Definition B.2.** A statistical manifold is represented by a parameter space $\mathcal{H}$ of a distribution $p(\mathbf{x} \mid \boldsymbol{\xi})$ and is endowed with the FIM as the Riemannian metric.

The Riemannian pullback metric is derived as in Eq. (24). Having defined the Riemannian pullback metric for VAEs with arbitrary likelihoods, one can extend the measurement of curve lengths in $\mathcal{Z}$ when mapped to $\mathcal{H}$ through $h$ as displayed by Eq. (26). This approach allows flexibility in the choice of the decoder, as long as the FIM of the chosen distribution type is tractable.

## C  BASELINE DESCRIPTION

### C.1  GAE

Here, we describe the Geodesic Autoencoder (GAE) from Huguet et al. (2022). For more details on the theoretical framework, we refer to the original publication. The GAE works by matching Euclidean distances between latent codes with the *diffusion geodesic distance*, which is an approximation of the diffusion ground distance in the observation space.

Briefly, the authors compute a graph with affinity matrix based on distances between observations $i$ and $j$ using a Gaussian kernel as:

$$(\mathbf{K}_\epsilon)_{ij} = k_\epsilon(\mathbf{x}_i, \mathbf{x}_j) , \tag{30}$$

with scale parameter $\epsilon$, where $\mathbf{x}_i, \mathbf{x}_j \in \mathcal{X}$ and $\mathcal{X}$ is the observation space. The affinity is then density-normalised by $\mathbf{M}_\epsilon = \mathbf{Q}^{-1} \mathbf{K}_\epsilon \mathbf{Q}^{-1}$, where $\mathbf{Q}$ is a diagonal matrix such that $\mathbf{Q}_{ii} = \sum_j (\mathbf{K}_\epsilon)_{ij}$. To compute the diffusion geodesic distance, the authors additionally calculate the diffusion matrix $\mathbf{P}_\epsilon = \mathbf{D}^{-1} \mathbf{M}_\epsilon$, with $\mathbf{D}_{ii} = \sum_{j=1}^n (\mathbf{M}_\epsilon)_{ij}$ and stationary distribution $\boldsymbol{\pi}_i = \mathbf{D}_{ii} / \sum_j \mathbf{D}_{jj}$. The diffusion geodesic distance between observations $\mathbf{x}_i$ and $\mathbf{x}_j$ is

$$G_\alpha(\mathbf{x}_i, \mathbf{x}_j) = \sum_{k=0}^{K} 2^{-(K-k)\alpha} \|(\mathbf{P}_\epsilon)_{i:}^{2^k} - (\mathbf{P}_\epsilon)_{j:}^{2^k}\|_1 + 2^{-(K+1)/2} \|\boldsymbol{\pi}_i - \boldsymbol{\pi}_j\|_1 , \tag{31}$$

with $\alpha \in (0, 1/2)$. The running value of $k$ in Eq. (31) defines the scales at which similarity between the random walks starting at $\mathbf{x}_i$ and $\mathbf{x}_j$ are computed.

Given the diffusion geodesic distance $G_\alpha$ defined in Eq. (31), the GAE model is trained such that the pairwise Euclidean distances between latent codes approximate the diffusion geodesic distances in

the observation space $\mathcal{X}$, in a batch of size $B$. Given an encoder $f : \mathbb{R}^G \to \mathbb{R}^d$, the reconstruction loss is optimised alongside a geodesic loss

$$\mathcal{L}_{\text{geodesic}} = \frac{2}{B} \sum_{i=1}^{N} \sum_{j>i} (\|f(\mathbf{x}_i) - f(\mathbf{x}_j)\|_2 - G_\alpha(\mathbf{x}_i, \mathbf{x}_j))^2 . \tag{32}$$

## C.2 ADDITIONAL COMPARISON BETWEEN FLATVI AND GAE

Although related in scope, FlatVI significantly differs from the Geodesic Autoencoder (GAE) proposed by Huguet et al. (2022). Firstly, GAE is a deterministic autoencoder optimised for reconstruction based on a Mean Squared Error (MSE) loss. As such, the model is not tailored to simulate gene counts. On the contrary, FlatVI optimises the decoder as an NB model, such that sampling from it produces discrete counts. This aspect has two advantages. By focusing on learning continuous parameters of a discrete likelihood, FlatVI explicitly models distributional properties of single-cell transcriptomics data, such as overdispersion, sparsity and discreteness. On the contrary, a fully connected Gaussian decoder produces dense and continuous cells, failing to preserve the characteristics of the data. Moreover, the GAE's regularisation relies on the construction of a k-nearest-neighbourhood to approximate the geodesic distance between data points. This method requires the computation of pairwise Euclidean distances in the observation space. As suggested previously, gene expression is high dimensional and, therefore, deceiving due to the Curse of Dimensionality. On the contrary, leveraging only the Jacobian and the output of the decoder to enforce latent space Euclideanicity, FlatVI is more suitable for larger datasets and eludes computing distances in high dimensions.

## D COMPUTATIONAL COMPLEXITY OF THE FIM COMPUTATION

Here is a breakdown of the complexity of Eq. (22):

- First, it is easier to reflect on the complexity of the computation from Eq. (7), $\mathrm{M}(\mathbf{z}) = \mathbb{J}_h(\mathbf{z})^T \mathrm{M}(\boldsymbol{\phi}) \mathbb{J}_h(\mathbf{z})$, which is the generalisation of Eq. (22).
- We call $G$ the number of genes and $d$ the dimensionality of the latent space. We also assume that $G >> d$. Since one mean parameter is decoded per gene dimension, $\phi$ is also $G$-dimensional.
- By the definition of decoder's Jacobian, $\mathbb{J}_h(\mathbf{z})$ is a $G \times d$ matrix. The Fisher information matrix ($\mathrm{M}(\boldsymbol{\phi})$) is a $G \times G$ since it represents the second derivative of the likelihood of a data point given the parameters. For a formal definition of the Fisher information metric, please refer to Eq. (16) and Eq. (22).
- Given this formulation, the complexity of $\mathbb{J}_h(\mathbf{z})^T \mathrm{M}(\boldsymbol{\phi})$ is $O(dG^2)$, yielding a $d \times G$ matrix. Such matrix is multiplied by a $G \times d$ matrix ($\mathbb{J}_h(\mathbf{z})$). This operation has complexity $O(d^2 G)$.
- Since $G >> d$, the complexity is bounded by the first product, thus it is $O(dG^2)$.

Note that the same result is obtained by evaluating the sum of matrices derived by outer products in Eq. (22) since the operation is equivalent. Finally, we only provided the computational complexity of the product evaluation. Bear in mind that in practice one must factor in the complexity of evaluating the decoder $h$ at the latent point $\mathbf{z}$.

## E MODEL SETUP

**Experimental details for Autoencoder models.** The Geodesic AE, NB-VAE and FlatVI models are trained via shallow 2-layer neural networks with hidden dimensions `[256, 10]`. Between consecutive layers, we include batch normalisation, as we found that it improves the reconstruction loss. Non-linearities are introduced by the `ELU` activation function. Models are trained for 1000 epochs monitoring the VAE loss for early stopping with a 20-epoch patience. The learning rate is set by default to `1e-3`. Additionally, we increase the KL divergence from 0 to 1 linearly across epochs in VAE models. The NB-VAE and FlatVI models are trained with a batch size of 32. We employed a batch size of 256 for the geodesic AE model after sweeping all values in $\{64, 100, 256\}$ and comparing the validation loss of different configurations. Importantly, the geodesic autoencoder

was trained to reconstruct $\log$-normalised counts, unlike the NB-VAE and FlatVI, since it does not assume an NB decoder. Finally, for training stability, all encoders are fed with $\log(1 + \mathbf{x})$ given an input $\mathbf{x}$. The list of hyperparameters explored for FlatVI together with the selected values based on the validation loss is provided in Tab. 3.

Table 3: Hyperparameter sweeps for training FlatVI. The hidden dimension column excludes the latent space layer, which is set to 10 unless specified otherwise. In bold, is the selected value used to present the results in the main.

|  | batch size | hidden dims | $\lambda$ |
|---|---|---|---|
| EB | **32**, 256, 512 | [1024, 512, 256], [512, 256], **[256]** | 0.001, 0.01, 0.1, **1**, 10 |
| Pancreas | **32**, 256, 512 | [1024, 512, 256], [512, 256], **[256]** | 0.001, 0.01, 0.1, **1**, 10 |
| MEF | **32**, 256, 512 | [1024, 512, 256], [512, 256], **[256]** | 0.001, 0.01, **0.1**, 1, 10 |

**Experimental details for OT-CFM.** For OT-CFM we use a 3-layer MLP with 64 hidden units per layer, a `SELU` activation function and a learning rate of `1e-3`. The velocity network is fed with a latent state concatenated with a scalar representing the time used for interpolation. Following suggestions from the OT-CFM repository [2], in each epoch we collect batches of cells from all time points to compute the objective for backpropagation. The variance hyperparameter $\sigma$ is set to 0.1 by default.

**The choice of the hyperparameter $\lambda$.** The hyperparameter $\lambda$ is crucial to control the degree of latent space flatness achieved by FlatVI (Tab. 1) but it can also lead to a decrease in the model likelihood if flattening is over-prioritised. A higher value of $\lambda$ corresponds to a more uniform (lower VoR) and flatter (lower CN) latent geometry. It is therefore important to choose a value of such a hyperparameter to ensure that flatness is enforced and reconstruction is good enough to predict realistic gene trajectories. In our experiments, we tested FlatVI followed by OT-CFM for different values of $\lambda$. Specifically, we increase the value of $\lambda$ as far as it produces an improved latent reconstruction by OT-CFM. For most real datasets, increasing $\lambda$ from 0.1 to 1 produces performance increases in trajectory reconstruction, whereas moving from 1 to 10 does not. For the more complex MEF dataset, moving from 0.1 to 1 did not improve OT-CFM-based trajectory reconstruction, so we decided to set $\lambda$ to 0.1.

## F    EVALUATION METRICS

### F.1    METRIC DESCRIPTION

**Condition number.** Given a metric tensor $\mathrm{M}(\mathbf{z})$, let $S_{\min}$ and $S_{\max}$ be its lowest and highest eigenvalues, respectively. The condition number (CN) is defined as the ratio

$$\mathrm{CN}(\mathrm{M}(\mathbf{z})) = \frac{S_{\max}}{S_{\min}} \; . \tag{33}$$

Notably, an identity matrix has a CN equal to 1. The CN is an indicator of the stability of the metric tensor. A well-conditioned metric with a CN close to 1 suggests that the lengths and angles induced by the metric are stable. A large condition number means that the distances are more stretched in some directions than others. On an Euclidean manifold with a scaled diagonal metric tensor, distances are preserved in all directions.

**Variance of the Riemannian metric.** In assessing the Riemannian metric, we introduce a key evaluation called the Variance of the Riemannian Metric (VoR) (Pennec et al., 2006). VoR is defined as the mean square distance between the Riemannian metric $\mathrm{M}(\mathbf{z})$ and its mean $\bar{\mathrm{M}} = \mathbb{E}_{\mathbf{z} \sim p_{\mathbf{z}}}[\mathrm{M}(\mathbf{z})]$. As suggested in Yonghyeon et al. (2021), we compute the VoR employing an affine-invariant Riemannian distance metric $d$, expressed as:

$$d^2(A, B) = \sum_{i=1}^{m} \left( \log \lambda_i(B^{-1}A) \right)^2 \; , \tag{34}$$

---

[2]`https://github.com/atong01/conditional-flow-matching`

where $\lambda_i(B^{-1}A)$ indicates the $i^{th}$ eigenvalue of the matrix $B^{-1}A$. VoR provides insights into how much the Riemannian metric varies spatially across different $\mathbf{z}$ values. When VoR is close to zero, it indicates that the metric remains constant throughout the support of $P_{\mathbf{z}}$. This evaluation procedure focuses solely on the spatial variability of the Riemannian metric and is an essential aspect of assessing the learned manifolds. Note that the expected value in Eq. (34) is estimated using batches of latent observations with size 256.

**Density.** (Naeem et al., 2020) Let $\mathbf{Y}$ and $\mathbf{X}$ be sets of generated and real data points with $M$ and $N$ observations, respectively. The neighbourhood sphere $B(\mathbf{x}_i, \mathrm{NND}_k(\mathbf{x}_i))$ is the spherical region around a real datapoint $\mathbf{x}_i$ with, as radius, the distance between $\mathbf{x}_i$ and the furthest of its k-nearest neighbours. For a generated sample $\mathbf{y}_i$, Density evaluates the number of real neighbourhood spheres that encompass $\mathbf{y}_j$. Mathematically, the metric is defined as:

$$\mathrm{Density}(\mathbf{X}, \mathbf{Y}) = \frac{1}{kM} \sum_{j=1}^{M} \sum_{i=1}^{N} \mathbb{1}_{\mathbf{y}_j \in B(\mathbf{x}_i, \mathrm{NND}_k(\mathbf{x}_i))} \ . \tag{35}$$

The Density metric rewards generated samples situated in regions where real samples are densely clustered. Importantly, Density can be higher than 1.

**Coverage.** (Naeem et al., 2020) Similar to Density, Coverage builds a k-nearest-neighbourhood around the real samples as the sphere $B(\mathbf{x}_i, \mathrm{NND}_k(\mathbf{x}_i))$. Given real and generated samples $\mathbf{X}$ and $\mathbf{Y}$ with $N$ and $M$ observations, Coverage is defined as:

$$\mathrm{Coverage}(\mathbf{X}, \mathbf{Y}) = \frac{1}{N} \sum_{i=1}^{N} \left( \mathbb{1}_{\exists j \ \mathrm{s.t.} \mathbf{y}_j \in B(\mathbf{x}_i, \mathrm{NND}_k(\mathbf{x}_i))} \right) \ . \tag{36}$$

Thus, Coverage measures the fraction of real samples whose neighbourhoods contain at least one generated sample. The score is bound between 0 and 1.

**Velocity Consistency.** (Gayoso et al., 2024) This metric quantifies the average Pearson correlation between the velocity $v(\mathbf{x}_j)$ of a reference cell $\mathbf{x}_j$ and the velocities of its neighbouring cells within the k-nearest-neighbour graph. It is mathematically expressed as:

$$c_j = \frac{1}{k} \sum_{\mathbf{x} \in \mathcal{N}_k(\mathbf{x}_j)} \mathrm{corr}(v(\mathbf{x}_j), v(\mathbf{x})) \ . \tag{37}$$

Here $c_j$ represents the Velocity Consistency, $k$ denotes the number of nearest neighbours considered in the k-nearest-neighbour graph, $\mathbf{x}_j$ is the reference cell, $\mathcal{N}_k(\mathbf{x}_j)$ represents the set of neighbouring cells. The value $\mathrm{corr}(v(\mathbf{x}_j), v(\mathbf{x}))$ is the Pearson correlation between the velocity of the reference cell $v(\mathbf{x}_j)$ and the velocity of each neighbouring cell $v(\mathbf{x})$. Higher values of $c_j$ indicate greater local consistency in velocity across the cell manifold.

### F.2 EXPERIMENT DESCRIPTION

**Simulation details.** We simulate 10-dimensional negative binomial data from three distinct categories parameterised by means following distinct distributions and the same inverse dispersion. The negative binomial means $\boldsymbol{\mu}$ are drawn from 10-dimensional Gaussian distributions with category-specific means -1,0 and 1. The inverse dispersion parameters $\boldsymbol{\theta}$ are again random and drawn from the same distribution across the different classes, namely a Gamma distribution with concentration equal to 2 and rate equal to 1. We exponentiate the means and take the absolute value of inverse dispersions to make them strictly positive. Note, that we do not use size factors in the simulation experiment. Overall, we simulate 1000 observations drawn uniformly from different categories.

**Parameter reconstruction and Euclidean distance approximation (Tab. 1).** We train FlatVI with different levels of regularisation strength for the flattening loss controlled via the $\lambda$ parameter. As evaluation metrics for the data reconstruction, we consider how well FlatVI reconstructs the mean $\boldsymbol{\mu}$ and inverse dispersion $\boldsymbol{\theta}$ parameters used to simulate individual cells. The last evaluation metric (indicated as $\mathrm{MSE}(\mathrm{Geo} - \mathrm{Euc})$) quantifies the MSE between geodesic and Euclidean distances evaluated across 1000 pairs of randomly sampled points. In short, 1000 couples of simulated cells are first sampled and encoded using FlatVI with different levels of regularisation. Between the elements $\mathbf{z}_1$ and $\mathbf{z}_2$ of each couple, we calculate:

- The Euclidean distance between the latent representations $\mathbf{z}_1$ and $\mathbf{z}_2$.
- The geodesic distance on the latent manifold $\mathcal{M}_{\mathcal{Z}}$ between $\mathbf{z}_1$ and $\mathbf{z}_2$. To compute the geodesic distance, we use the StochMan software [3] Given a metric tensor, the geodesic distance between two points on a manifold can be computed as the length of the shortest curve connecting such two points along the manifold, following Eq. (5). We approximate the shortest connecting curve with a cubic spline learnt to solve the minimisation task in Eq. (5), hence the curve that leads to the shortest walk on the manifold between the two points. Thanks to the relationship between the Fisher information (Eq. (16)) and the KL divergence between infinitesimal displacements on the latent manifold (see Eq. (29)), we reformulate the task as follows:

$$d_{\text{latent}}(\mathbf{z}_1, \mathbf{z}_2) = \inf_{\gamma(t)} \int_0^1 \text{KL}(p(\mathbf{x}|h(\gamma(t))), p(\mathbf{x}|h(\gamma(t + \mathrm{d}t)))) \mathrm{d}t \,, \tag{38}$$

$$\text{where} \quad \gamma(0) = \mathbf{z}_1, \ \gamma(1) = \mathbf{z}_2 \,.$$

Here, $p(\mathbf{x}|h(\gamma(t)))$ is a negative binomial distribution conditioned on the parameters derived as the decoded latent interpolation point. Note that $t$ here refers to the interpolation. In practice, we optimise the curve in Eq. (38) as a spline $\hat{\gamma}(t)$ defined over 100 discretised interpolation steps.

Both Euclidean and geodesic distances result in two vectors of 1000 values (one per couple), which we compare to each other via MSE. A lower MSE signifies that geodesic distances better approximate Euclidean distances. All these results were computed over three training repetitions of models with distinct regularisation strengths.

**Spearman correlation between Euclidean and geodesic distances and neighbourhood overlap metrics (Tab. 4).** We additionally assess to what extent the Euclidean distances and the pullback geodesics induce a comparable neighbourhood structure as well as to what extent these distances are correlated. Across 5 repetitions, we sample 50 simulated points, encode them with VAEs trained with different regularisation strengths and compute:

- The geodesic distance between all pairs of sampled observations using Eq. (38).
- The Euclidean distance between all pairs of sampled observations.

Here, we only use 50 observations per repetition due to the computational burden of Eq. (38). After we collect the Euclidean and geodesic distances we calculate their average Spearman correlation per data point. Moreover, using the two pairwise distance matrices, we obtain two distinct neighbourhood structures (we evaluate the 3 and 5 nearest neighbours to account for local structure). Given the neighbourhoods from the Euclidean and pullback geodesic distances, we calculate the neighbourhood overlap metric as the average proportion of nearest neighbours assigned to a data point both according to geodesics and Euclidean distances. A high value of such a metric signifies that, on average, the neighbourhood structure according to the pullback geodesic distance corresponds to the Euclidean one.

**Riemannian metrics and path visualisation.** Riemannian metrics only take the metric tensor at individual latent codes as input (see App. F.1). The metric tensor for individual observations is calculated following Eq. (13). The geodesic paths in Fig. 2**c** are again approximated by cubic splines optimising the task in Eq. (38). We compute such paths for 10 randomly drawn pairs in the region exhibiting high VoR in the unregularised VAE model for demonstration purposes.

**Trajectory reconstruction experiments.** In Table 2, we explore the performance of OT on different embeddings based on the reconstruction of held-out time points. For the EB dataset, we evaluate the leaveout performance on all intermediate time points. Conversely, on the MEF reprogramming dataset (Schiebinger et al., 2019) we conduct our evaluation holding out time points 2, 5, 10, 15, 20, 25 and 30 to limit the computational burden of the experiment. Note that the Pancreas dataset used for Fig. 3 could not be used for this analysis, since it only has two time points: an initial and a terminal one. After training OT-CFM excluding the hold-out time point $t$, we collect the latent representations of cells at $t - 1$ and simulate their trajectory until time $t$, where we compare the generated cells with

---

[3] `https://github.com/MachineLearningLifeScience/stochman/tree/black-box-random-geometry`

the ground truth via distribution matching metrics both in the latent and in the decoded space ($L^2$ and 2-Wasserstein distance in the latent space, Density, Coverage, 2- Wasserstein distance and MMD in the decoded space). Density and Coverage in the gene expression space are used to evaluate the mixing between the real and generated cells. Their values are computed in the PCA space of 50 dimensions of the $\log$-transformed real and generated gene counts, considering 10 neighbours. For the latent reconstruction quantification, generated latent cell distributions at time $t$ are standardised with the mean and standard deviation of the latent codes of real cells at time $t$ to make the results comparable across embedding models. Results in Tab. 2 are averaged across three seeds.

**Fate mapping with CellRank.** We first train representation learning models on the Pancreas dataset. Then, following the setting proposed by Eyring et al. (2022), we learn a velocity field over the latent representations of cells by matching time points 14.5 to 15.5 with OT-CFM and input the velocities to CellRank (Lange et al., 2022; Weiler et al., 2023). Using the function `g.compute_macrostates(n_states, cluster_key)` of the GPPCA estimator for macrostate identification Reuter et al. (2019), we look for 10 to 20 macrostates. If OT-CFM cannot find one of the 6 terminal states within 20 macrostates for a certain representation, we mark the terminal state as missed (see Figure 3). Terminal states are computed with the function `compute_terminal_states(method, n_states)`. Velocity consistency is estimated using the scVelo package (Bergen et al., 2020) through the function `scv.tl.velocity_confidence(adata_latent_flat)`. The value is then averaged across cells.

## G   DATA

### G.1   DATA DESCRIPTION

**Embryoid Body (EB).** Moon et al. (2019) measured the expression of 18,203 differentiating human embryoid single cells across 5 time points. From an initial population of stem cells, approximately four lineages emerged, including Neural Crest, Mesoderm, Neuroectoderm and Endoderm cells. Here, we resort to a reduced feature space of 1241 highly variable genes. OT has been readily applied to the embryoid body datasets in multiple scenarios (Tong et al., 2020; 2023), making it a solid benchmark for time-resolved single-cell trajectory inference. The data is split into 80% training and 20% test sets.

**Pancreatic Endocrinogeneris (Pancreas).** We consider 16,206 cells from Bastidas-Ponce et al. (2019) measured across 2 time points corresponding to embryonic days 14.5 and 15.5. In the dataset, multipotent cells differentiate branching into endocrine and non-endocrine lineages until reaching 6 terminal states. Challenges concerning such dataset include bifurcation and unbalancedness of cell state distributions across time points (Eyring et al., 2022). The data is split into 80% training and 20% test sets.

**Reprogramming Dataset (MEF).** We consider the dataset introduced in Schiebinger et al. (2019), which studies the reprogramming of Mouse Embryonic Fibroblasts (MEF) into induced Pluripotent Stem Cells (iPSC). The dataset consists of 165,892 cells profiled across 39 time points and 7 cell states. For this dataset, we keep 1479 highly variable genes. Due to its number of cells, such a dataset is the most complicated to model among the considered. The data was split into 80% training and 20% test sets.

### G.2   DATA PREPROCESSING

We use the Scanpy (Wolf et al., 2018) package for single-cell data preprocessing. The general pipeline involves normalisation via `sc.pp.normalize_total`, log-transformation via `sc.pp.log1p` and highly-variable gene selection using `sc.pp.highly_variable_genes`. 50-dimensional embeddings are then computed via PCA through `sc.pp.pca`. We then use the PCA representation to compute the 30-nearest-neighbour graphs around single observations and use them for learning 2D UMAP embeddings of the data. For the latter steps, we employ the Scanpy functions `sc.pp.neighbours(adata)` and `sc.tl.umap(adata)`. Raw counts are preserved in `adata.layers["X_counts"]` to train FlatVI.

### G.3 DETAILS ABOUT COMPUTATIONAL RESOURCES

Our model is implemented in Python 3.10, and for deep learning models, we used PyTorch 2.0. For the implementation of NeuralODE-based simulations, we use the torchdyn package. Our experiments ran on different GPU servers with varying specifications: GPU: 16x Tesla V100 GPUs (32GB RAM per card) / GPU: 2x Tesla V100 GPUs (16GB RAM per card) / GPU: 8x A100-SXM4 GPUs (40GB RAM per card)

## H ALGORITHM

**Problem setting.** For time-resolved scRNA-seq data, cells are collected in $T$ unpaired distributions $\{\nu_t\}_{t=0}^T$. Individual time points correspond to separate snapshot datasets $\{\mathbf{X}_t\}_{t=0}^T$, each with $N_t$ observations. Every snapshot is mapped to tuples $\{(\mathbf{Z}_t, \mathbf{l}_t)\}_{t=0}^T$ of latent representations $\mathbf{Z}_t \in \mathbb{R}^{N_t \times d}$ and size factors $\mathbf{l}_t \in \mathbb{N}_0^{N_t}$ following the setting described in Sec. 3.1. We wish to learn the dynamics of the system through a parameterised function in the latent space $\mathcal{Z}$ of a VAE, taking advantage of its continuity and lower dimensionality properties.

**Size factor treatment.** Since the size factors $\mathbf{l}_t$ required for decoding are observed variables derived from single-cell counts in the dataset, their values are not available when simulating novel cell trajectories from $t = 0$, hindering the use of the decoder to recover individual gene evolution. Assuming that the size factor is a real number and related to the cell state, we include $\log l_t$ in the latent dynamics and infer its trajectory together with the latent state representation $\mathbf{z}_t$. The $\log$ is taken for training stability. Therefore, we learn a velocity field $v_\xi : [0, 1] \times \mathbb{R}^{d+1} \to \mathbb{R}^{d+1}$ on the concatenated state $\mathbf{s}_t = [\mathbf{z}_t, \log l_t]$. The time-resolved vector field $v_\xi$ is modelled by matching subsequent pairs of cell distributions.

**Gene expression trajectories.** If the latent space $\mathcal{Z}$ can be injectively mapped to the parameter manifold $\mathcal{H}$, trajectories in $\mathcal{Z}$ correspond to walks across the continuous parameter space via the stochastic decoder $h$. The temporal trajectory of the likelihood parameter vector $\boldsymbol{\mu}_t$ is given by

$$\boldsymbol{\mu}_t = h\left(\mathbf{s}_0 + \int_0^t v_\xi(t', \mathbf{s}_{t'})\, \mathrm{d}t'\right) , \tag{39}$$

where $\boldsymbol{\mu}_0 = h(\mathbf{s}_0)$ and we express the decoder function $h(\mathbf{z}_t, l_t)$ from Eq. (3) as $h(\mathbf{s}_t)$ for simplicity. Then, assuming a gene-wise constant inverse dispersion $\boldsymbol{\theta}_g$, discrete trajectories of gene counts follow the noise model $\mathbf{x}_t \sim \mathrm{NB}(\boldsymbol{\mu}_t, \boldsymbol{\theta})$.

---

**Algorithm 1** Train FlatVI

---

**Require:** Data matrix $\mathbf{X} \in \mathbb{N}_0^{N \times G}$, batch size $B$, maximum iterations $n_{\max}$, encoder $f_\psi$, decoder $h_\phi$, flatness loss scale $\lambda$
**Ensure:** Trained encoder $f_\psi$, decoder $h_\phi$, and inverse dispersion parameter $\boldsymbol{\theta}$
  Randomly initialize gene-wise inverse dispersion $\boldsymbol{\theta}$
  Randomly initialize the identity matrix scale $\alpha$ as a trainable parameter
  **for** $i = 1$ **to** $n_{\max}$ **do**
    Sample batch $\mathbf{X}^b \leftarrow \{\mathbf{x}_1, ..., \mathbf{x}_B\}$ from $\mathbf{X}$
    $\mathbf{l}^b \leftarrow \texttt{compute\_size\_factor}(\mathbf{X}^b)$
    $\mathbf{Z}^b \leftarrow f_\psi(1 + \log \mathbf{X}^b)$
    $\boldsymbol{\mu} \leftarrow h_\phi(\mathbf{Z}^b, \mathbf{l}^b)$
    $\mathcal{L}_{\mathrm{KL}} \leftarrow \texttt{compute\_kl\_loss}(\mathbf{Z}^b)$
    $\mathcal{L}_{\mathrm{recon}} \leftarrow \texttt{compute\_nb\_likelihood}(\mathbf{X}^b, \boldsymbol{\mu}, \boldsymbol{\theta})$
    $\mathrm{M}(\mathbf{Z}^b) \leftarrow$ Eq. (22)
    $\mathcal{L}_{\mathrm{flat}} \leftarrow \texttt{MSE}(\mathrm{M}(\mathbf{Z}^b), \alpha \mathbb{I}_d)$
    $\mathcal{L} = \mathcal{L}_{\mathrm{recon}} + \mathcal{L}_{\mathrm{KL}} + \lambda \mathcal{L}_{\mathrm{flat}}$
    Update parameters via gradient descent
  **end for**

---

---

**Algorithm 2** Train latent OT-CFM with FlatVI

---

**Require:** Datasets $\{\mathbf{X}_t\}_{t=0}^T$, variance $\sigma$, batch size $B$, initial velocity function $v_\xi$, maximum iterations $n_{\max}$, trained encoder $f_\psi$

**Ensure:** Trained velocity function $v_\xi$

    $\{\mathbf{Z}_t\}_{t=0}^T \leftarrow f_\psi(\{1 + \log \mathbf{X}_t\}_{t=0}^T)$

    $\{\mathbf{l}_t\}_{t=0}^T \leftarrow \texttt{compute\_size\_factor}(\{\mathbf{X}_t\}_{t=0}^T)$

    $\{\mathbf{S}_t\}_{t=0}^T \leftarrow \texttt{timewise\_concatenate}(\{\mathbf{Z}_t\}_{t=0}^T, \{\mathbf{l}_t\}_{t=0}^T)$

    **for** $i = 1$ **to** $n_{\max}$ **do**

        Initialize empty array of velocity predictions $\mathbf{V}$

        Initialize empty array of velocity ground truth $\mathbf{U}$

        **for** $t_{\text{traj}} = 0$ **to** $T - 1$ **do**

            Randomly sample batches with $B$ observations $\mathbf{S}_{t_{\text{traj}}}^b, \mathbf{S}_{t_{\text{traj}}+1}^b$

            $\pi \leftarrow \text{OT}(\mathbf{S}_{t_{\text{traj}}}^b, \mathbf{S}_{t_{\text{traj}}+1}^b)$

            $(\mathbf{S}_{t_{\text{traj}}}^b, \mathbf{S}_{t_{\text{traj}}+1}^b) \sim \pi$

            $t \sim \mathcal{U}(0, 1)$

            $\mathbf{S}^b \leftarrow \mathcal{N}(t\mathbf{S}_{t_{\text{traj}}}^b + (1-t)\mathbf{S}_{t_{\text{traj}}+1}^b, \sigma^2 \mathbb{I}_d)$

            Append $v_\xi(t + t_{\text{traj}}, \mathbf{S}^b)$ to $\mathbf{V}$

            Append $(\mathbf{S}_{t_{\text{traj}}+1}^b - \mathbf{S}_{t_{\text{traj}}}^b)$ to $\mathbf{U}$

        **end for**

        $\mathcal{L}_{\text{OT-CFM}} \leftarrow \|\mathbf{V} - \mathbf{U}\|^2$

        Update parameters via gradient descent

    **end for**

---

# I    ADDITIONAL RESULTS

## I.1    SIMULATED NEGATIVE BINOMIAL DATA

### I.1.1    SIMULATED DATA VISUALISATION

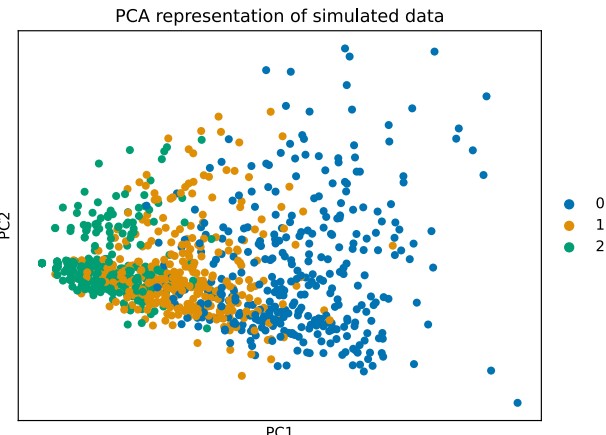

Figure 5: The PCA dimensionality reduction plot of the simulated data from three categories.

### I.1.2    COMPARISON WITH EUCLIDEAN SPACE METRICS

In Fig. 6 we provide an extension to Fig. 2 where we add how the VoR, CN and geodesic paths should appear in the Euclidean space. Notably, both CN and VoR are uniform, equating to 1 and 0, respectively. Furthermore, while some points in the simulated data exhibit high values for VoR and CN, the representation from FlatVI is more compatible with the expected one under Euclidean

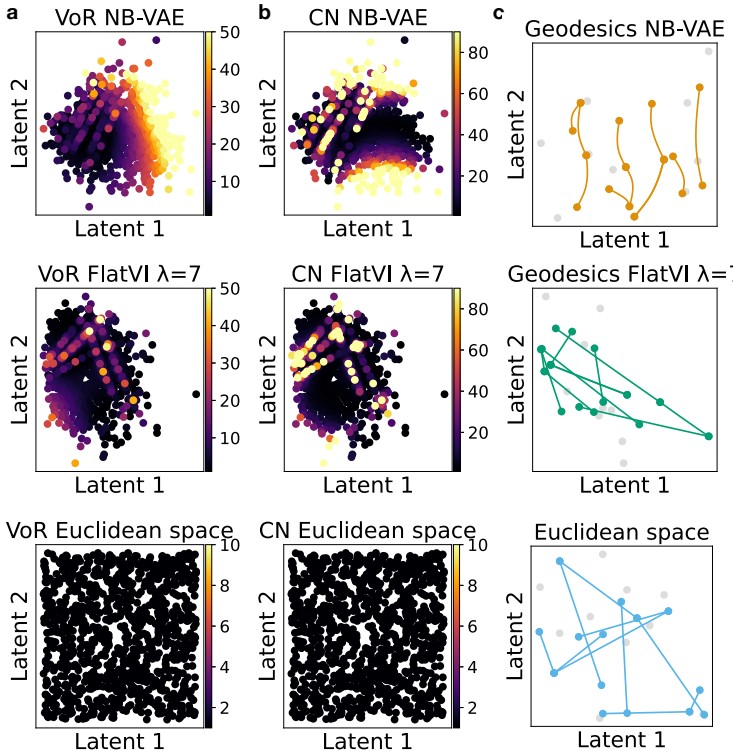

Figure 6: Comparison between the latent geometries of the NB-VAE (top row), FlatVI trained with $\lambda = 7$ (middle row) and an example Euclidean space of 1000 points evaluated in terms of **(a)** Variance of the Riemannian metric (VoR), **(b)** Condition Number (CN) and **(c)** Straightness of the geodesic paths connecting pairs of latent points. The Euclidean panels are simulated as uniformly sampled points on a regular grid.

geometry than a normal NB-VAE. Additionally, path straightness is better preserved in FlatVI's latent geodesics compared to the counterpart, validating the purpose of our model.

### I.1.3 ADDITIONAL METRICS

In addition to the metrics reported in Tab. 1, in Tab. 4 we provide further results that justify the usage of our flattening loss in the simulated data setting. The new metrics (Spearman correlation coefficient and Neighbourhood overlap) are described in App. F.2. Such values represent how much the latent pullback geodesic and Euclidean distances and the neighbourhood structures deriving from them correspond. As can be inferred from the table, adding the regularisation promotes an overall improvement of the metrics. While Spearman correlation does not drastically separate results, adding the flattening regularisation marks an improvement in neighbourhood preservation metrics of up to 18% when considering 3 neighbours and 14% when using 5-point neighbourhoods. In other words, the pullback geodesics is better reflected by the Euclidean distance when applying our regularisation, which provides evidence of the working principles of our flattening loss.

### I.1.4 ANALYSIS OF SUB-OPTIMALLY FLATTENED REGIONS

In Fig. 2, we show that introducing our flattening loss component in the VAE model training ensures a lower and more uniform Riemannian metric throughout the space, as well as lower CN. However, some points of our simulation dataset still display high decoding distortion and variance in the Riemannian metric as signals of insufficient flattening. In Fig. 7, we show that latent paths between points of high VoR and their neighbours do display some curvature, indicating regions of the manifold with sub-optimal Euclidanisation. We investigated what causes points to exhibit a high VoR in both FlatVI ($\lambda$=7) and the regular NB-VAE. First, we found that regions of high VoR and CN in FlatVI tend to overlap, while in NB-VAE they are less correlated (see Fig. 8a). Hence, while for most of

Table 4: Comparison between FlatVI and the unregularised NB-VAE ($\lambda = 0$). Spearman (Geo-Euc) represents the Spearman correlation between latent Euclidean distances, where the latter are computed using Eq. (5). The Neighborhood metrics (respectively accounting for 5 and 3 neighbours) measure the proportion of neighbours per data point that correspond between the Euclidean and Geodesic distance matrices. The error bars are derived from 5 repetitions of the experiment. Every repetition consists of drawing 50 generated data points, computing the pairwise geodesic and Euclidean latent distances and deriving the above metrics.

| $\lambda$ | Spearman (Geo-Euc) ($\uparrow$) | Neighbourhood overlap (5nn) ($\uparrow$) | Neighbourhood overlap (3nn) ($\uparrow$) |
|---|---|---|---|
| $\lambda = 0$ | $0.94_{\pm 0.00}$ | $0.50_{\pm 0.01}$ | $0.66_{\pm 0.00}$ |
| $\lambda = 1$ | $0.95_{\pm 0.00}$ | $0.57_{\pm 0.01}$ | $0.63_{\pm 0.00}$ |
| $\lambda = 3$ | $\mathbf{0.96_{\pm 0.01}}$ | $\mathbf{0.68_{\pm 0.03}}$ | $0.77_{\pm 0.00}$ |
| $\lambda = 5$ | $\mathbf{0.97_{\pm 0.01}}$ | $0.58_{\pm 0.01}$ | $0.67_{\pm 0.01}$ |
| $\lambda = 7$ | $\mathbf{0.97_{\pm 0.01}}$ | $0.60_{\pm 0.01}$ | $0.72_{\pm 0.01}$ |
| $\lambda = 10$ | $\mathbf{0.97_{\pm 0.00}}$ | $\mathbf{0.68_{\pm 0.02}}$ | $\mathbf{0.80_{\pm 0.03}}$ |

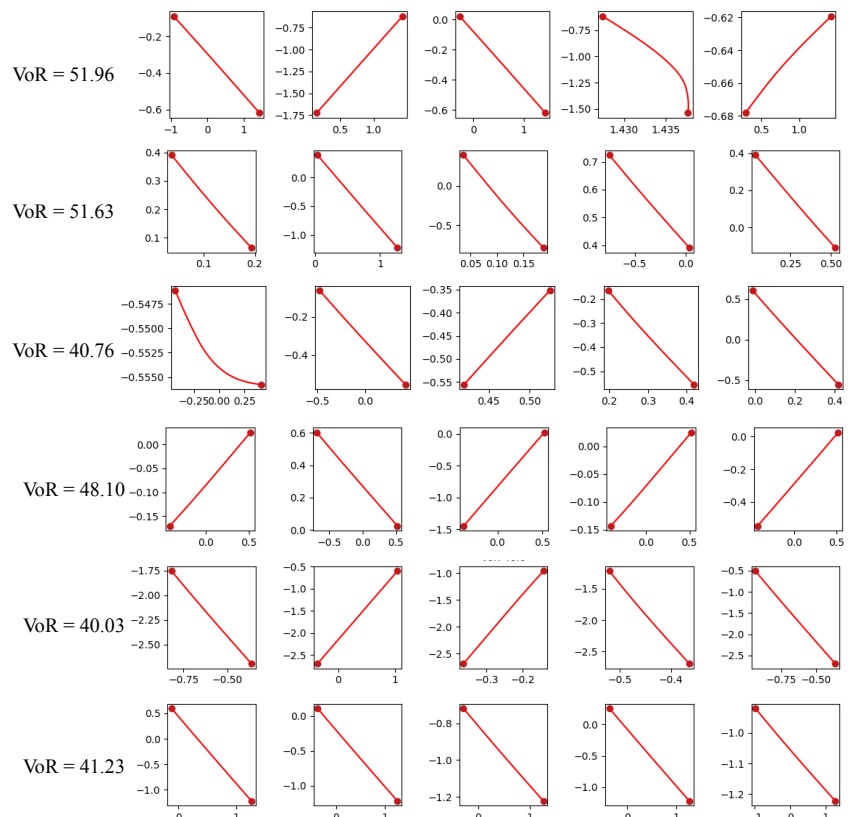

Figure 7: Geodesic paths between regions with high VoR and CN in the FlatVI embeddings with $\lambda = 7$. Every row represents a point with high VoR and CN. The columns are five randomly sampled neighbours in the dataset. Every plot represents the geodesic path between the point with high VoR and CN and the associated neighbour. As a representation of high VoR and CN, we select points with the VoR value larger than 30.

the observations, flattening applies, the pullback metric from the decoder violates uniformity and preservation of angles and distances in some portion of the space.

We check the label annotation of the insufficiently flattened regions by overlying their VoR and CN values onto the UMAP plot of the real data (see Fig. 8b-c). By this analysis, we note that the high VoR and CN data points are concentrated at the inter-class boundaries in FlatVI's latent space (see Fig. 9). In other words, observations in regions of the manifold enriched by different classes representing state transitions are more likely to fail to flatten. The fact that high VoR and CN are concentrated in

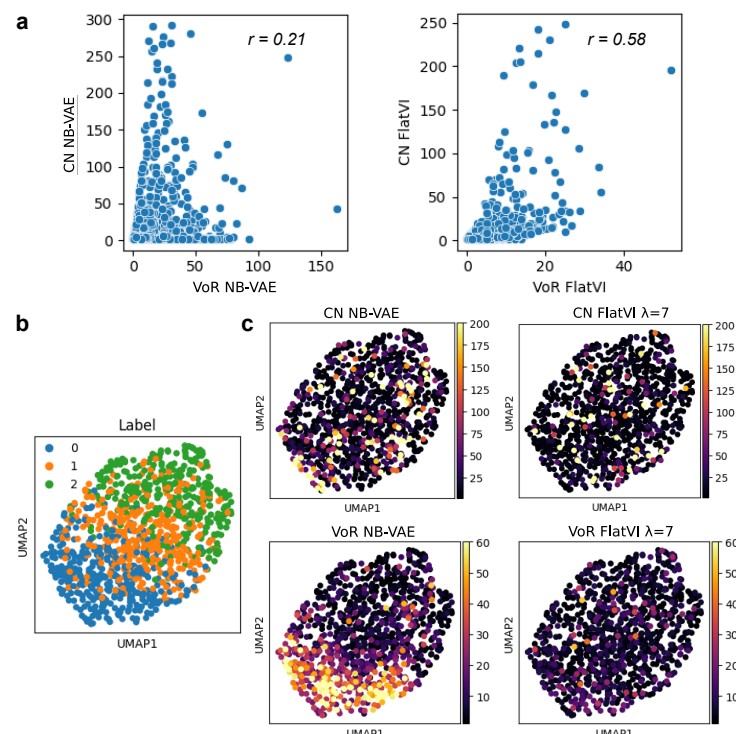

Figure 8: (**a**) The scatter plot and correlation values between VoR and CN in the embeddings computed by NB-VAE and FlatVI. (**b-c**) The UMAP plots of the real data coloured by class and CN and VoR values from the NB-VAE and FlatVI embeddings.

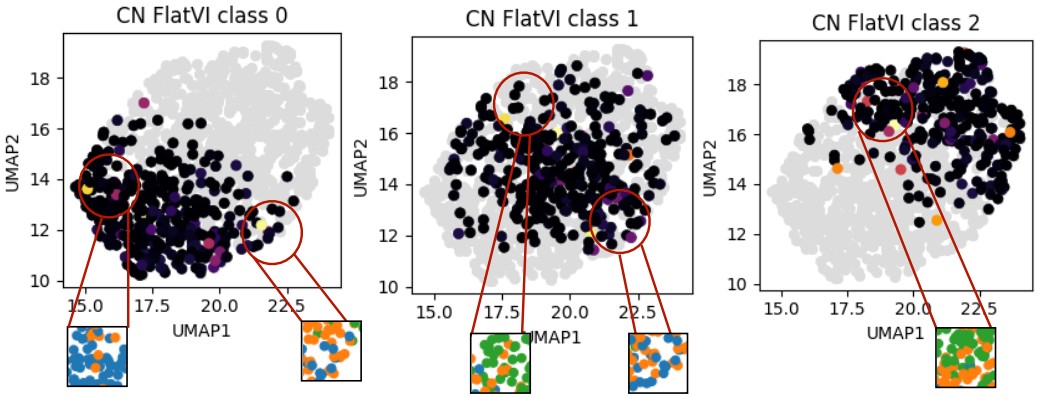

Figure 9: The UMAP plots computed on the actual simulated count data coloured by the CN from the FlatVI embeddings. Highlighted are regions with high VoR and CN. External boxes represent the label compositions of the highlighted regions (for a label-specific colour legend, see Fig. 8)b.

regions of class heterogeneity may suggest that FlatVI fails to unfold some fast-changing manifold regions at the intersection between classes and the decoder needs to violate the isometry between the Euclidean latent space and the statistical manifold to ensure a proper reconstruction.

## I.2  CIRCULAR MANIFOLD SIMULATION

Inspired by a toy example involving noisy circular data (Arvanitidis et al., 2021), we designed an experiment where data points are sampled from a 2D circular manifold embedded within a 3D

Gaussian statistical manifold with constant variance ($\sigma^2 = 0.1$). The 3D manifold is endowed with the Fisher Information metric for the Gaussian distribution, as described in Arvanitidis et al. (2021).

In this experiment, we train a 3D Variational Autoencoder (VAE) with flattening regularisation strengths $\lambda \in \{0, 0.5, 1\}$ and a 2D latent space. After training, linear interpolations between random pairs of points in the latent space are computed, and the resulting geodesic paths are plotted on the 2D manifold. The results, shown in Fig. 10, display red paths between point pairs, representing the decoded means of the Gaussian likelihood.

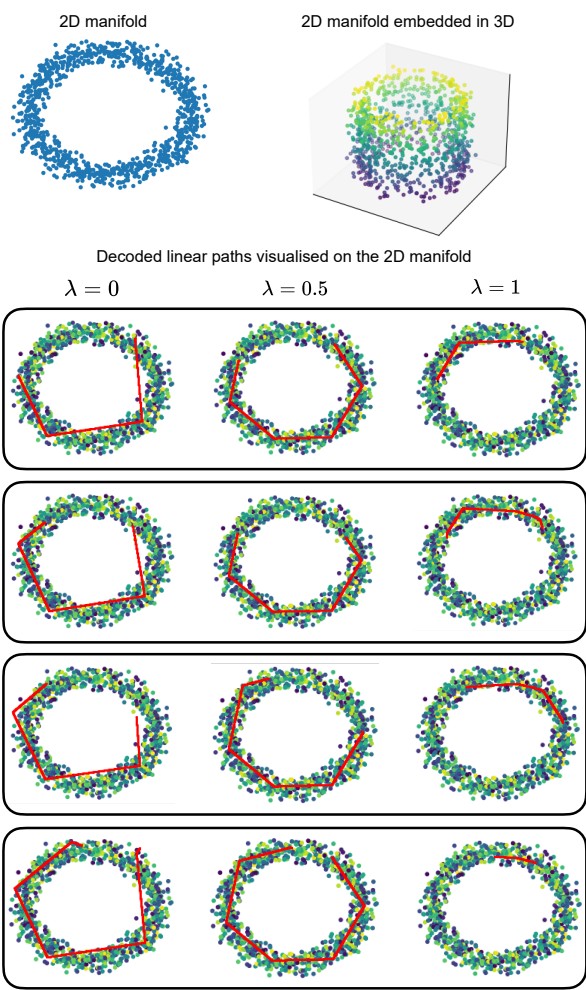

Figure 10: **Top.** A 2D circular manifold embedded in a 3D Gaussian statistical manifold. **b.** Manifold interpolations (red lines) between pairs of points computed by training a Gaussian VAE with varying levels of flattening regularisation. Paths are obtained by performing linear interpolations between the latent codes of the source and target points. These linear interpolations are decoded, and the mean of the resulting decoded Gaussian distributions is visualised on the 2D manifold.

### I.3 VELOCITY ESTIMATION

We compared our whole pipeline involving the combination of FlatVI and OT-CFM with the scVelo (Bergen et al., 2020) and veloVI (Gayoso et al., 2024) models for RNA velocity analysis. Fig. 11 and Tab. 5 show examples of how our approach favourably compares with velocity estimation methods, namely inferring a proper velocity field in Acinar cells and detecting all terminal states with high velocity consistency in the Pancreas dataset.

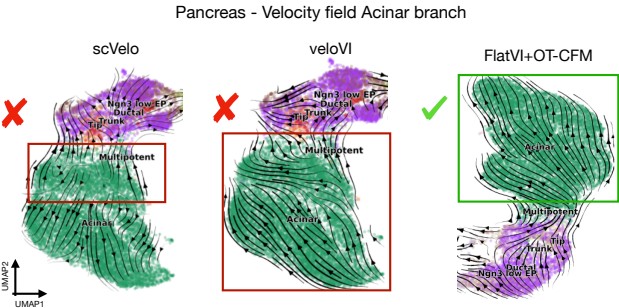

Figure 11: Comparison between the vector field learnt on the Acinar branch of the Pancreas dataset by using OT-CFM in combination with FlatVI and standard RNA velocity algorithms.

Table 5: Number of terminal states computed by CellRank and velocity consistency using the representations and velocities learnt by FlatVI+OT-CFM and standard RNA velocity algorithms.

| Method | Terminal states | Consistency |
|---|---|---|
| FlatVI + OT-CFM | **6** | **0.94** |
| veloVI | 5 | 0.92 |
| scVelo | 4 | 0.80 |

### I.4 TRAJECTORY VISUALISATION OF REAL DATA

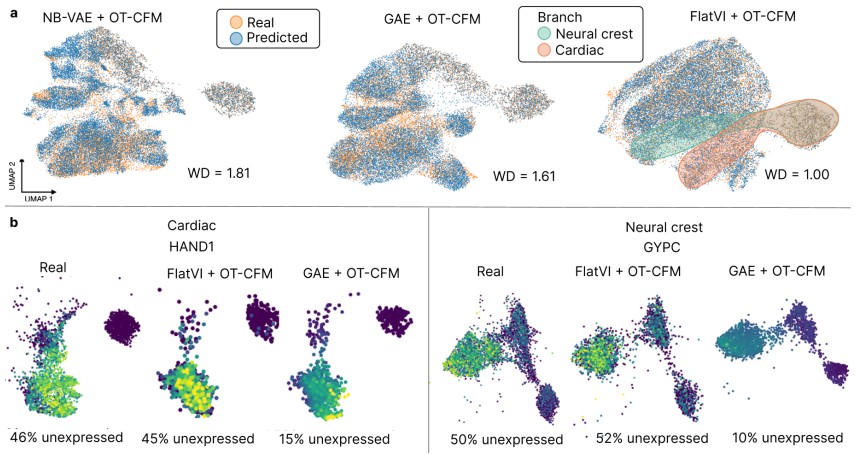

Figure 12: Prediction of scRNA-seq in time. (**a**) Overlap between real and simulated latent samples in the EB dataset. WD indicates the 2-Wasserstein distance between real and generated latent cell representations. (**b**) 2D UMAP plots of real and predicted cell counts from the cardiac and neural crest lineages of the EB dataset, comparing FlatVI and GAE as representations for OT-CFM. Colours indicate the predicted $\log$ gene expression of the reported lineage drivers GYPC and HAND1. Under each UMAP plot, we calculate the percentage of unexpressed marker instances along the trajectory.

## I.5 ADDITIONAL TABLES

Table 6: Runtime, in seconds, evaluated over a single forward pass considering different batch sizes for each compared model. The runtime is tested over 10 repetitions using random inputs with 2k dimensions.

| | Runtime (s) | | |
|---|---|---|---|
| Batch size | GAE | NB-VAE | FlatVI |
| 8 | $0.119_{\pm 0.012}$ | $0.000_{\pm 0.000}$ | $0.009_{\pm 0.004}$ |
| 16 | $0.095_{\pm 0.012}$ | $0.001_{\pm 0.000}$ | $0.007_{\pm 0.002}$ |
| 32 | $0.115_{\pm 0.012}$ | $0.001_{\pm 0.000}$ | $0.004_{\pm 0.003}$ |
| 64 | $0.099_{\pm 0.002}$ | $0.001_{\pm 0.000}$ | $0.005_{\pm 0.000}$ |
| 128 | $0.116_{\pm 0.007}$ | $0.002_{\pm 0.000}$ | $0.009_{\pm 0.004}$ |
| 256 | $0.218_{\pm 0.017}$ | $0.002_{\pm 0.000}$ | $0.009_{\pm 0.002}$ |
| 512 | $0.513_{\pm 0.021}$ | $0.002_{\pm 0.000}$ | $0.015_{\pm 0.004}$ |
| 1024 | $1.522_{\pm 0.013}$ | $0.003_{\pm 0.000}$ | $0.015_{\pm 0.001}$ |

Table 7: Separation between initial and terminal lineage states evaluated in terms of clustering metrics in the latent spaces of the distinct models. Different representation spaces are compared on how well they unroll developmental trajectories.

| | Silhouette Score (↑) | | | Calinski-Harabasz (↑) | | | Davies-Bouldin (↓) | | |
|---|---|---|---|---|---|---|---|---|---|
| | EB | Pancreas | MEF | EB | Pancreas | MEF | EB | Pancreas | MEF |
| GAE | **0.28** | 0.15 | 0.09 | **1608.56** | 1723.13 | 11232.84 | **1.03** | 1.50 | 2.99 |
| NB-VAE | 0.19 | 0.26 | 0.21 | 940.87 | 2191.48 | 19440.38 | 1.28 | 1.56 | 2.35 |
| FlatVI | 0.21 | **0.50** | **0.31** | 983.41 | **6986.31** | **45372.75** | 1.18 | **0.73** | **1.50** |

Table 8: Univariate Wasserstein-2 distance between simulated and real marker gene expression for different lineage branches of the EB dataset across models. A lower value indicates that the model better approximates marker gene trajectories along the branch.

| | Wasserstein-2 real-simulated markers (↓) | | | | | | | | | | | |
|---|---|---|---|---|---|---|---|---|---|---|---|---|
| | Cardiac | | | Neural Crest | | | Endoderm | | | Neuronal | | |
| | GATA6 | HAND1 | TNN2 | NGFR | GYPC | PDGFRB | SOX17 | GATA3 | CDX2 | LMX1A | ISL1 | CXCR4 |
| GAE | 0.17 | 0.28 | 0.23 | 0.05 | 0.24 | 0.07 | 0.24 | 0.14 | 0.11 | 0.04 | 0.04 | 0.18 |
| NB-VAE | 0.03 | 0.24 | 0.07 | 0.07 | 0.09 | 0.04 | **0.08** | 0.07 | **0.02** | 0.02 | 0.06 | 0.07 |
| FlatVI | **0.02** | **0.09** | **0.03** | **0.02** | **0.05** | **0.03** | **0.08** | **0.02** | **0.02** | **0.01** | **0.02** | **0.03** |

