# OpenReview forum: "Enforcing Latent Euclidean Geometry in VAEs for Statistical Manifold Interpolation"
_ICLR.cc/2025/Conference — Submitted to ICLR 2025_

### Official Review · Reviewer_7m6g · 2024-10-21

**Soundness:** 2
**Presentation:** 2
**Contribution:** 2
**Rating:** 5
**Confidence:** 2

**Summary:**

The authors consider the approximation of a manifold by a VAE. Here, they introduce a new regulariser for the training loss which enforces that geodesics on the learned manifold correspond to straight lines in the latent space. They pay particular attention to an application with cellular data from computational biology.

**Strengths:**

The regulariser of the loss function is well-motivated by the pull-back of the Fisher-Information-Geometry. At the same time it is simple to implement and seems to be effective. Also the application looks sensible, but I have to admit that it is a bit out of my own expertise (therefore the low confidence of the review).

**Weaknesses:**

During reading the paper I came across a couple of confusions and comments, which I list below. Summarising, I think that this paper needs some improvements about its assumptions and the clarity of writing. In the case that the authors improve the paper during the rebuttal, I am willing to reconsider my evaluation.

## Confusion about the kind of manifolds which is considered

The authors mention at several points of the paper that they consider manifolds "with complex geometry". The authors should specify what is meant by this term. For instance, there is no chance to approximate a manifold with holes (like in Fig 1) with a single latent space, since there exists no homeomorphism between two spaces of different topology.

As a simple example in intuitive language: when we can consider the $\mathbb{S}^1$ embedded in the $\mathbb{R}^2$ and approximate it by a VAE with one-dimensional latent space, we have to "wrap the line around the circle". This leads to a point, where "the two ends of the line meet". In particular, there exist points which are very close in the data space, but far away from each other in the latent space.

In particular, in the general case, geodesics might not exists or might not be unique. In both cases, the representation with a flat latent space is not possible.

## Evaluation of the flattening loss

The authors should evaluate the proposed flatting loss in a lighter setting, where the results can easily be visualised. One suggestion for such an experiment could be the following:
Take a two dimensional exapmle manifold embedded in the $\mathbb{R}^3$ (e.g., the torus, sphere, swiss roll, etc.) and learn a VAE with different flattening loss regularisation strengths (and the Riemannian metric inherited from the space where the manifold is embedded in). Then, take some points on the manifold located opposite of each other, connect them by a straight line in the latent space and plot the resulting path on the manifold. If the flattening loss works properly, these lines should become more regular for stronger regularisation.

## Clarity of the model

The paper could benifit a lot from a complete and formal description of the model somewhere in the beginning, answering the following questions: What kind of data is given? From where to where maps the decoder (so what is the manifold and how does it relate to the data)? Where comes the time dynamics into play?

The description in Section 3.1 is insufficient. For example, I came across the following confusions:

- please define $\mathcal X$ and $\mathcal Z$.

- $\phi$ denotes at the same time a variable from the parameters space of $\mathbb{P}$ and the paramteres of the decoder. This is not a very good style. In particular, in (2) there appears the expression $p_\phi(x|h_\phi(z))$. Inserting the definition of $p_\phi$ from (1), this is equal to $\mathbb{P}(x|h_\phi(h_\phi(z)))$, which is most certainly not, what the authors meant.

Even though these errors are minor and most readers should be able to figure out what is meant, they make the paper really hard to read.


## Minor comments

- The notations from Sec 3.2 require that the decoder is a diffeomorphism between the latent space and its image. In this setting this corresponds to the property that $\mathbb{J}_h(z)$ is full-rank for all $z$ from the latent space. While I think that it is common to ignore this assumption for numerical computations, this should be mentioned.

- In Section 3.2: It is a bit odd to cite recent research papers for classical concepts for differential geometry. It would be better to refer to some general text book (like these research papers do).

- eqt (11): please specify the used norm.

- Fig 1: Why does the visualisation of the VAE interpolation map to a point outside the learned manifold?

- Probably evaluating the flattening loss is slower than the usual VAE loss since it requires differentiating twice insted of once through the decoder (once for computing the FIM and once for taking the gradient). Is that correct? If yes, this should be mentioned somewhere (even though I don't believe that this limitation is too important).

- line 323: Please specify the parameter space ($\theta,\mu>0$?)

**Questions:**

I state already plenty of questions in the weaknesses part. Additionally, I wondered how the approach scales with the dimension of the manifold, and in particular with the dimension of the space where it is embedded in. Is the approach feasible for high-dimensional manifolds? I guess that at some points the matrix-vector multiplication for the Gram matrices become expansive. Which dimension range is mainly targeted by the authors?

---

> ### Author Response · Authors · 2024-11-22
>
> We thank the reviewer for their detailed feedback and suggestions, which improved the quality of the manuscript. Below, we provide detailed answers to the raised questions and concerns.
>
> > Confusion about manifold
>
> Thank you for your feedback. We acknowledge that referring to the manifold of interest as a "complex data manifold" is a coarse and under-elaborated description. Here, we take the opportunity to clarify what we aim to convey instead.
>
> One important aspect of our model is its application to experimental data, which often includes noise, non-continuous domains, and lacks a clear association with a geometry that has closed-form geodesics. This is the primary reason we leverage the latent space of a VAE for manifold learning and regularise it towards a simple geometric structure to facilitate interpolation. In essence, the main desideratum of our model is to provide a framework for efficiently learning how to interpolate the experimental data manifold. Unlike well-defined geometries, such as that of a sphere where geodesic paths are straightforward to compute, experimental data manifolds typically do not allow for direct geodesic traversal. This data-oriented approach has been considered quite often recently due to its relevance in applied fields [1,2].
>
> Since our primary focus is single-cell RNA-seq, we specifically design our method around the concept of a statistical manifold—i.e., a manifold of probability distributions from a particular family. This concept is well-suited to data regimes where an accurate noise model can be assumed and captured by the decoder of a VAE, such as the negative binomial distribution for scRNA-seq. In this context, we leverage information geometry to connect the geometry of the latent space manifold with that of the decoded space, governed by the Fisher Information Metric.
>
> We have revised the text to better reflect our purpose, replacing the term **complex manifold** with the idea of **experimental data manifold** or, in more specific passes, with a specific reference to the **statistical manifold spanned by the VAE deocder**. Examples of such clarifications are in the abstact (L15) and in the caption of Fig.1. Additionally, we updated Fig.1 by removing the hole to avoid potential misconceptions and to provide a more general visualisation of the concept of a data manifold. We also updated the introduction (see lines 54–64) to more accurately convey our modelling objectives and to explain how our framework optimises towards them.
>
> We hope the reviewer finds our elaboration and the changes made to the text insightful.
>
> > Evaluation of the flattening loss in a lighter setting
>
> We thank the reviewer for their relevant suggestion. Inspired by the toy example in [3], where the authors consider noisy circular data, we designed an experiment simulating data points sampled from a 2D circular manifold embedded within a 3D Gaussian statistical manifold with constant variance. The 3D manifold is equipped with the Fisher Information metric for the Gaussian distribution, as described in [3].
>
> In this experiment, we trained a 2D VAE with regularisation strengths $\lambda \in \{0, 0.5, 1\}$, as suggested by the reviewer. After training, we computed linear paths between random pairs of points in the latent space and plotted the resulting geodesic interpolations on the 2D manifold.
>
> The results are shown in Fig. 10 (Section I.2), where the red paths between point pairs represent the decoded means of the Gaussian likelihood. Analysing the results, we observed several interesting patterns that support the working principles of our model. Without regularisation ($\lambda = 0$), decoded geodesic paths deviate from the manifold and fail to take the shortest route along it between source and target points. With $\lambda = 0.5$, the adherence to the manifold improves, but interpolations still follow longer routes between points. Only with $\lambda = 1$ does the regularisation ensure that geodesic paths remain on the manifold and correspond to the shortest path between points.
>
> These results highlight the effectiveness of our approach, with the empirical findings providing support of our model assumptions and design.

---

> > ### Author Response · Authors · 2024-11-22
> >
> > > Clarifications
> >
> > We thank the reviewer for pointing out the missing information. We made the following changes.
> >
> > - Definition of $\mathcal{X}$ and $\mathcal{Z}$
> >     - We have added proper descriptions for the space of observations $\mathcal{X}=\mathbb{N}_0^G$ and the latent space $\mathcal{Z}=\mathbb{R}^d$
> > - Overloading of phi variable
> >     - We, furthermore, addressed the overloading of the $\phi$-variable. Specifically, we updated L145 and L161 and removed the $h_{\phi}$ from Eq.2 to remove ambiguity.
> >
> >
> > > Clarity of the model
> >
> > * Section 3.1 now addresses the concerns raised regarding the type of input, as it provides:
> >     * A description of the data type, specifically high-dimensional count data characterised by sparsity and overdispersion.
> >     * Details on the dimensionality of the input.
> >     * An explanation of the encoding and decoding mechanisms employed.
> >     * A breakdown of all components of the negative binomial loss.
> >
> > * Section 4.1 formalises the problem of learning population dynamics in detail and justifies the use of optimal transport for this task. In single-cell temporal modelling, it is common to collect snapshots of cellular datasets measured at different experimental timepoints and to learn continuous models that approximate the evolution of the cellular population over time. Each timepoint is associated with a dataset of single-cell data, which we embed using FlatVI. The representation space produced by our model serves as a better input for Euclidean optimal transport, as it is regularised to approximate a Euclidean manifold.
> >
> >
> > * Section 4.3 explains how the regularisation framework, described in general terms in the previous sections, can be formulated for a negative binomial decoder modelling single-cell data.
> >
> >
> > > Minor comments
> >
> > We thank the reviewer for thoroughly reviewing the manuscript and providing such detailed feedback. Specifically:
> > - We added a note regarding the full rank assumption of the Jacobian in L193.
> > - We updated our references in Sec 3.2. to properly account for the general mathematical concepts that we are using.
> > - We signified the L2-norm in Eq.11.
> > - Q: Why does normal VAE map outside of manifold?
> >     - We visualize the standard NB-VAE outside the manifold, as such a model is likely to deviate from the manifold when using (linear) latent interpolations to deduce biological processes. In contrast, FlatVI remains more faithful to the true geodesics.
> > - We included a note regarding the increased runtime in the Remarks paragraph of Section 4.2. Additionally, we direct the reviewer to our complexity estimate for computing the FIM (Appendix D) and the runtime comparisons with the considered baselines (Table 6).
> > - We added the specification regarding the parameters of the negative binomial distribution $\mu$ and $\theta$ in L330.
> >
> >
> > > Scaling behaviour
> >
> > * As expected, adding an extra term to regularise the pullback metric tensor leads to increased model runtime, as it requires evaluating the Jacobian of the decoder ($\mathbb{J}_h$ in the paper, where $h$ is the decoder function).
> > * In this work, we focus on tabular data with around 2k dimensions, which we select as the most variable gene features in single-cell datasets. We embed the data into the latent space using MLP models with one to three layers of non-linearities. Despite the added regularisation overhead, a single forward pass is completed in a reasonable amount of time (see Tab. 6) and training runtime is contained. Of course, moving to larger input spaces (e.g, images) would require more careful considerations. Furthermore, our Jacobian-regularised model scales more efficiently than GAE, which requires calculating pairwise distances between input observations to estimate the diffusion geodesic distance in the data domain.
> > * To improve the efficiency of computing the pullback metric, we use the Jacobian-vector product (JVP) rather than constructing the full Jacobian. For the complete metric tensor, we apply JVP to calculate directional derivatives along each standard basis vector, dynamically assembling the necessary parts of the decoder's Jacobian. This approach eliminates the computational and memory costs associated with forming the full Jacobian, while still providing the necessary terms for the metric calculation.
> >
> >
> > We thank the reviewer once again for their time and remain available for further clarifications.
> >
> > [1] Kapuśniak, Kacper, et al. "Metric flow matching for smooth interpolations on the data manifold." arXiv preprint arXiv:2405.14780 (2024).
> >
> > [2] de Kruiff, Friso, et al. "Pullback Flow Matching on Data Manifolds." arXiv preprint arXiv:2410.04543 (2024).
> >
> > [3] Arvanitidis, Georgios, et al. *"Pulling back information geometry."* arXiv preprint arXiv:2106.05367 (2021).

---

> > > ### Comment · Reviewer_7m6g · 2024-11-25
> > >
> > > Thank you very much for your response. I appreciate the effort the authors have put into revising the manuscript. After reading over it, I have decided to maintain my score for the following reasons:
> > >
> > > While I acknowledge that the authors addressed several minor comments, there are still some aspects where the work could benefit from further clarity. For instance, the rephrasing of "complex manifold" to "experimental data manifold" remains unclear, and I think that a formal statement is necessary, which properties of the manifold are required such that the proposed method is applicable.
> > >
> > > Additionally, I found some of the results in Fig. 10 to be strange:
> > >
> > > - The piecewise constant nature of the trajectories without flattening loss was unexpected. I had anticipated a more irregular behavior.
> > > - The example highlights the significance of data manifold topology: with the presence of a hole, it is impossible to approximate the manifold with a single decoder. There (provably) exist points which are close on the manifold but very far away in the latent space.
> > > - I do not see any reason why the flattening loss should help to find the shorter direction around a circle. To my understanding, the flattening loss enforces that the decoder is locally an isometry (whether this is even possible is not discussed in the paper) such that geodesics remain geodesics. However, on a circle both ways around are geodesics and a local property (like the Gram matrix which is used in the flattening loss) cannot prefer one of these trajectories. In fact, one can here directly see the problem from my previous point. Since geodesics on the manifold are not unique but geodesics in the latent space are unique, even the strongest flattening loss cannot provide a one-to-one relation between them.
> > >
> > > I also share the viewpoint of reviewer tvQk regarding the flattening loss being a limited (and fully empirical) contribution, particularly since the manuscript does not consider important theoretical questions (e.g. does an embedding from $\mathbb{R}^d$ to a $d$-dimensional manifold maintaining geodesics exist?). However, as already written in my initial review, I cannot say how much of a contribution is added by the biological application (which is not my field, I leave this to the other reviewers and area chair).

---

> > > > ### Author Response · Authors · 2024-11-25
> > > > **Follow-up on Rebuttal Responses**
> > > >
> > > > Dear Reviewer 7m6g,
> > > >
> > > > Thank you once again for taking the time to review our work. We greatly value your detailed feedback and recognize your critique as an opportunity to refine and enhance our presentation.
> > > >
> > > > In response to the additional concerns you raised, we would like to provide further justification for the significance and relevance of our work.
> > > >
> > > > > The employed type of manifold.
> > > >
> > > > The concept of the manifold of experimental data is used only at the beginning before introducing the idea of a statistical manifold spanned by the decoder of the variational autoencoder. In the rest of the paper, we hint that the type of Riemannian manifold we deal with is a statistical manifold of probability distributions given by the type of likelihood model implemented by the decoder. In sections 3.1 and 3.2 we provide an extensive description of this concept, together with the definition of the Fisher Information Metric (FIM) as the metric defining the geometry of the manifold.
> > > >
> > > > While we understand we do not provide specific assumptions concerning the exact topology of the manifold, we believe it is inaccurate to state that our description of the geometric setting is limited to the concept of *complex manifold*. Our outline of the setting follows an established formalisation by existing work [1,2], where the authors elaborate on a connection between auteoncoders and Riemannian manifolds.
> > > >
> > > > > Application perspective and contribution.
> > > >
> > > > FlatVI has an application focus, therefore it is directed to the physical sciences track. We kindly disagree with the statement that our regularisation applied to an empirical setting represents a minor contribution, as it is the first time someone has tried to establish geometric constraints on the stochastic decoder of single-cell VAEs for downstream applications. While the end product might appear simple from a methodological perspective, it builds on a novel intuition and formulation of cellular data as a statistical manifold, which produces empirical evidence of improved performance on OT-based tasks assuming Euclidean space.
> > > >
> > > > > Limits of the flattening approach.
> > > >
> > > > We do not claim that the proposed method achieves perfect Euclidean geometry in the latent space. Instead, our goal is to make the latent space geometry more closely resemble Euclidean geometry by adapting the decoder to map straight lines in the latent space to non-linear geodesics in the reconstruction space, at least within a local neighbourhood of the data (see the answer to reviewer L8EZ, point 2 of the **communication** section). We argue that this adjustment is sufficient to enhance the performance of downstream tasks, which depends on having straight-line distances in the latent space.
> > > >
> > > > > Simulations
> > > >
> > > > Notice that in all path examples we provided, we do not choose two opposite ends of the circle but specific points where the minor arc corresponds to the geodesics.
> > > >
> > > > In summary, we address the problem of manifold learning from a data-driven perspective, as seen in related work [3,4], and we would be very grateful if the reviewer could take this aspect into account in their assessment.
> > > >
> > > > Thank you again for your time and consideration.
> > > >
> > > > Best regards,
> > > >
> > > > The Authors
> > > >
> > > > [1] Arvanitidis, Georgios, Søren Hauberg, and Bernhard Schölkopf. "Geometrically enriched latent spaces." arXiv preprint arXiv:2008.00565 (2020).
> > > >
> > > > [2] Arvanitidis, Georgios, et al. "Pulling back information geometry." arXiv preprint arXiv:2106.05367 (2021).
> > > >
> > > > [3] Huguet, Guillaume, et al. "Manifold interpolating optimal-transport flows for trajectory inference." Advances in neural information processing systems 35 (2022): 29705-29718.
> > > >
> > > > [4] Kapuśniak, Kacper, et al. "Metric flow matching for smooth interpolations on the data manifold." arXiv preprint arXiv:2405.14780 (2024).

---

### Official Review · Reviewer_L8EZ · 2024-10-29

**Soundness:** 4
**Presentation:** 4
**Contribution:** 3
**Rating:** 6
**Confidence:** 4

**Summary:**

The paper proposes a regularization of variational autoencoders (VAEs) such that the natural representation geometry is flattened. This makes shortest paths (geodesics) near-linear, such that optimal transport dynamics are more readily applicable (OT dynamics often rely on straight-line interpolants). The work is demonstrated in the single-cell regime and results are promising.

**Strengths:**

* The paper provides a sound and sensible combination of techniques from VAEs, differential geometry, and optimal transport.
* The proposed model allows for more reliable data interpretations, which is important e.g. in the studied life science domains (but also in other domains).
* The paper is easy to follow and generally well-written.

**Weaknesses:**

## Communication
* It is not always entirely clear what the actual contribution of the paper is and what was previously known. For example, the introduction states that "In particular, no prior work has extended existing frameworks for learning latent geodesic trajectories to statistical manifolds of
discrete probability distributions", but this is exactly what the "Pulling back Information geometry" paper did. The present paper has significant novelty over the "Pulling back information geometry" paper, but the communication becomes unclear in terms of what constitutes the contribution.
* The paper is often phrased to indicate that the fitted model will have an Euclidean latent geometry. This is obviously incorrect. What the paper actually does is regularize towards Euclidean geometry. This is merely a matter of phrasing things differently, but I would appreciate more clarity here.

## Computations
* As far as I can tell, differentiating through the regularizer (Eq. 11) will be a bit expensive as it requires higher-order derivatives of the decoder.

**Questions:**

* The "Pulling back Information Geometry" was able to avoid differentiating through the metric by instead approximating it locally with the KL. Is something similar feasible to make optimizing Eq 11 cheaper?
* Will code be made available?
* A big part of prior work on latent representation geometries focuses on how model uncertainty seems to reflect the topology of the data manifold. As far as I can tell, the presented approach is not influenced by model uncertainty. Is this correctly understood? If so, would it be difficult to incorporate uncertainty?

---

> ### Author Response · Authors · 2024-11-22
>
> We thank the reviewer for their thorough and constructive feedback. We appreciate the positive recognition of our technical approach, as well as the highlighting of our model's potential for reliable data interpretations in domains such as the life sciences. Below, we provide detailed answers to the raised questions and concerns.
>
> > Better communication towards our contribution
> - We thank the reviewer for raising this point. In the updated version of our manuscript, we tried to delineate better between known results and our contribution towards aligning latent space interpolations with geodesics on statistical manifolds. Our primary focus was the introduction, where the reviewer detected ambiguity in the aspects that distinguish our framework from prior work. We reformulate lines 54 to 64 by listing a series of desiderata for an ideal latent manifold interpolation model, and explaining why existing work does not exhaustively fulfill them all. Right after this paragraph, FlatVI is introduced as a method incorporating all desirable aspects. We believe the reviewer's suggestion significantly helped us improve the flow of our introduction.
> - Furthermore, we discuss our contribution in detail in our response to reviewer tvQk.
> - We also updated our manuscript to reflect that our methods regularises towards / approximates a Euclidean geometry. See, for example, L22, L83, L418, or L529 (all such changes are highlighted in blue in the text). We appreciate the reviewer's feedback and believe that these changes contributed towards the clarity of our work.
>
> > Computational complexity
>
> * Expectedly, introducing an additional term to regularise the pullback metric tensor causes an increase in the model's runtime, as it involves the evaluation of the Jacobian of the decoder ($\mathbb{J}_h$ in the paper, where $h$ is the decoder function).
> * In practice, this work focuses on tabular data, which we embed into the latent space using MLP models with one to three layers of non linearities. Although the regularisation introduces a computational overhead, a single forward step is completed in a reasonable time (see Tab. 6). Moreover, in the considered setting, our Jacobian-based regularised model scales better than GAE, which requires the computation of pairwise distances between input observations to estimate the diffusion geodesic distance in the data domain.
> * To compute the pullback metric more efficiently, we use the Jacobian-vector product (JVP) rather than constructing the full Jacobian. For the full metric tensor, we apply JVP to compute directional derivatives along each standard basis vector, dynamically assembling the necessary components of the decoder's Jacobian. This approach avoids the computational and memory overhead of explicitly forming the full Jacobian while providing the required terms for the metric computation.
>
> > Q: Is a KL-approximation also possible in our setting?
>
> This is a very interesting question. To clarify, given a latent code $\mathbf{z}$ and an observation $\mathbf{x}$, the authors in [2] define an approximation for the elements of the metric tensor $\mathbf{M}$ as a function of the KL divergence between the likelihoods $p(\mathbf{x}|h(\mathbf{z}))$ and $p(\mathbf{x}|h(\mathbf{z}+\delta \mathbf{z}))$, where $\delta$ represents a perturbation of $\mathbf{z}$ and $h$ is a decoder function. Initially, we considered applying this scheme for computational efficiency. However, it's important to recall our likelihood model:
>
> $$ \mathbf{x} \sim \mathrm{NB}(\mu, \theta) $$
>
> $$ \mu = l \cdot \mathrm{softmax}(h_{\phi}(\mathbf{z})) $$
>
> where $l = \sum_g \mathbf{x_g}$ and $h_\phi$ is a neural network that decodes gene proportions (indicated as $\rho_\phi$ in the manuscript).
>
> When implementing the approximate metric method, we encountered an issue: Decoding perturbations of $\mathbf{z}$ introduced some ambiguity in how to handle the size factor. Since $l$ is not a function of $\mathbf{z}$, it was unclear what value to use for this variable when computing the perturbed likelihood $p(\mathbf{x}|h(\mathbf{z} + \delta \mathbf{z}))$, which is required for the metric approximation. As a result, we decided to adhere to the standard version of the metric tensor, which provides a closed-form expression in the case of the negative binomial distribution.
>
> > Q: Will the code be made available
>
> Yes, we will make the code available as part of the final submission. The software release will follow the structure of scVI tools [1] to promote a faster introduction of the model to the community.

---

> > ### Comment · Reviewer_L8EZ · 2024-11-26
> > **Thanks for the detailed reply**
> >
> > I appreciate the author's replies, which matched my expectations.
> >
> > For now, I will leave my score unchanged, but will revisit this during discussions with the other reviewers.

---

> ### Author Response · Authors · 2024-11-22
>
> > Q: Is there a connection to model uncertainty?
>
> Unlike FlatVI, the authors in [2] do not regularise the latent manifold to a simple geometry. Consequently, to perform geodesic interpolations between latent codes $\mathbf{z}_1$ and $\mathbf{z}_2$, they must learn and optimise a cubic spline that connects the two points based on energy minimisation. To ensure the fitted curve does not model paths that leave the data's support, they introduce an uncertainty regularisation term. This term extrapolates to high uncertainty if the curve represents a trajectory incompatible with the data support.
>
> In contrast, FlatVI does not require learning parameterised latent curves via cubic splines as in [2]. This is because we regularise the latent pullback geometry to approximate a Euclidean manifold, where geodesics between $\mathbf{z}_1$ and $\mathbf{z}_2$ are simple linear paths that map to data geodesics via the decoder. Ideally, when Euclidean geometry is well-approximated and geodesics are accurately decoded along the data manifold, our model remains uninfluenced by uncertainty.
>
>
> Nevertheless, implementing uncertainty regularisation in our setting is still feasible, particularly in scenarios where FlatVI is underregularised and the latent space is not fully flattened. As the reviewer suggested, our loss *pushes* the model towards a latent Euclidean geometry but does not necessarily guarantee it, especially depending on the regularisation strength. If regularisation is incomplete, one might opt to optimise geodesic curves instead of relying on linear paths to trace interpolations between $\mathbf{z}_1$ and $\mathbf{z}_2$. In such cases, it would be necessary to define a notion of maximum uncertainty for the negative binomial likelihood spanned by the decoder under mean/inverse dispersion regularisation. For instance, we could set the inverse dispersion parameter of the distribution to infinity when modelling transitions outside the data's support, as the variance of the negative binomial distribution is given by $ \mu + \theta \mu^2$, where $\theta$ is the inverse dispersion parameter. This strategy could help enforce meaningful transitions while preserving compatibility with the underlying data structure. Adding such an extension to FlatVI would be an interesting avenue for the future.
>
>
> We once again thank the reviewer for their insightful suggestions and hope that we were able to address all remaining concerns satisfactorily. We look forward to further discussion.
>
> [1] Gayoso, Adam, et al. "A Python library for probabilistic analysis of single-cell omics data." Nature biotechnology 40.2 (2022): 163-166.
>
> [2] Arvanitidis, Georgios, et al. *"Pulling back information geometry."* arXiv preprint arXiv:2106.05367 (2021).

---

### Official Review · Reviewer_tvQk · 2024-11-03

**Soundness:** 4
**Presentation:** 4
**Contribution:** 2
**Rating:** 5
**Confidence:** 2

**Summary:**

This paper generalizes the Riemannian structure for the statistical manifold with Fisher Information Metric (FIM) to a flattened version and solves OT-CFM problem. More specifically, it proposes a regularizer $\mathcal{L}_{flat}$ which penalizes the curvature from $M$.

**Strengths:**

The paper shows improvement in many applications, including simulated data and real biological problems. Good performance is shown.

**Weaknesses:**

My current understanding is: the main contribution is only adding a regularizer $||M-I||$. I would think it is an incremental contribution, since the definition of ELBO and $M(z)$ are both known. I understand this forces the ELBO loss to become Euclidean loss, but I need some help understanding the main contribution better.

Could you also elaborate on the novelty and significance of applying this technique specifically to statistical manifolds and single-cell RNA sequencing data, please?

**Questions:**

Could you please explicitly state which parts of Sections 3 and 4 are novel contributions versus background or existing work? I am not an expert but it seems to me that all the contents in Sec 3 and 4 are existing results except Eq. 11 and 12. It would be helpful if you could highlight the key innovations other than the regularization term.

Tiny typo: there is an extra comma in line 259

---

> ### Author Response · Authors · 2024-11-22
>
> We thank the reviewer for taking the time to review our work and for providing the opportunity to elaborate further on our contribution. Below, we highlight our contribution in greater detail.
>
> > The biological contribution
>
> * scRNA-seq profiles high-dimensional states, with each feature representing a gene's expression as a discrete count of RNA molecules. The data is inherently sparse and skewed due to high dropout rates and overdispersion caused by experimental inaccuracies and biological variability.
>
> * These characteristics necessitate modeling noise with appropriate discrete models. VAEs are ideal, as their decoder maps the latent space $\mathbf{z}$ to the parameters of an arbitrary likelihood model. For scRNA-seq, a negative binomial decoder is most suitable, as it models counts while capturing the mean-variance trend beyond Poisson assumptions. Crucially, decoding $\mathbf{z}$ represents a cell as the parameters of a negative binomial distribution, effectively making each **decoded cell a probability distribution.**
>
> * In the VAE literature, the decoder is often assumed to span a statistical manifold of probability distributions from a specific family. Similarly, we can interpret decoded single cells as points on such a manifold. **To the best of our knowledge, no prior work has approached manifold learning assuming that single-cell data lie on the statistical manifold of negative binomial distributions defined by the NB-VAE decoder.**
>
> * Why take this approach? Many scRNA-seq tasks rely on interpolations in a latent or reduced-dimensional space. **Our work explores whether we can push such interpolations to meaningful paths on the statistical manifold spanned by the decoder**. Since the negative binomial model is the most empirically accurate for scRNA-seq noise, we aim for latent interpolations to align with the geometry of this statistical manifold.
>
> > The machine learning contribution
>
> - To connect with the application setting, we develop an approach to approximate geodesic walks on the decoded manifold as trajectories on a simpler latent manifold geometry. Although geodesic interpolation methods exist, we aim for the following properties:
>     - **Interpolation of empirical data manifolds:** The approach should interpolate an empirical data manifold where no closed-form geodesic distance between two points exists. This distinguishes our framework from methods studying interpolations on standard manifolds with tractable geodesics (e.g., Riemannian Flow Matching [1] or Fisher Flow Matching [2]).
>     - **Cost-efficient latent manifold interpolation:** Interpolations along the latent manifold should be computationally inexpensive. Without regularising the latent manifold to a simpler geometry, interpolations are typically parameterised geodesic curves requiring costly simulations. For instance, Arvaniditis et al. [3] learn cubic splines that minimise the geodesic distance between two latent points using the pullback metric $M(\mathbf{z})$ (see Eq. 5 in our paper). Computing such interpolations involves iterative updates to the spline's weights, resulting in an expensive optimisation problem.
>     - **Suitability for complex, non-continuous data domains:** The strategy must handle challenging domains, such as scRNA-seq data. While methods like Metric Flow Matching [4] address some of these challenges, their manifold regularisation technique relies on a continuous data space. Indeed, the authors primarily consider low-dimensional approximations of gene expression vectors, which is often insufficient for capturing biological variability.
>     - **Real-world applicability:** In the spirit of the *Applications to physical sciences* track, the framework should connect to existing tools to encourage its adoption in practical scenarios.
>
>
> * FlatVI achieves all of these goals:
>     - It builds on the pullback geometry of VAEs to enable latent manifold learning with real high-dimensional data.
>     - It introduces a regularisation strategy that encourages linear latent interpolations to approximate geodesic paths on the statistical manifold defined by the decoder. **Notably, no prior method has explored flattening a VAE's latent space using the Fisher-based pullback metric**.
>     - It is applied to the complex scRNA-seq domain, formalising for the first time the connection between the geometry of latent cellular representations and the stochastic negative binomial manifold of the decoder. This framework is also generalisable to other count-based distributions, such as the Poisson.

---

> > ### Author Response · Authors · 2024-11-22
> >
> > >  Note on the loss.
> >
> > Note that the purpose of our formulation is not to transform the ELBO into a Euclidean loss. Instead, the regularisation we introduce constrains the decoder to approximate the latent space as a Euclidean space, where the shortest paths between points are straight lines. The decoder then maps these straight-line paths to geodesic interpolations on the statistical manifold. The theoretical foundations of our framework are detailed in Appendix A and B.
> >
> >
> > We sincerely thank the reviewer for their feedback. We hope that our explanations and the updated manuscript address the reviewer's concerns and provide more clarity. We are happy to answer further questions during the discussion period.
> >
> >
> > [1] Chen, Ricky TQ, and Yaron Lipman. "Riemannian flow matching on general geometries." arXiv preprint arXiv:2302.03660 (2023).
> > [2] Davis, Oscar, et al. "Fisher flow matching for generative modeling over discrete data." arXiv preprint arXiv:2405.14664 (2024).
> > [3] Arvanitidis, Georgios, et al. *"Pulling back information geometry."* arXiv preprint arXiv:2106.05367 (2021).
> > [4] Kapuśniak, Kacper, et al. "Metric flow matching for smooth interpolations on the data manifold." arXiv preprint arXiv:2405.14780 (2024).

---

> ### Comment · Reviewer_tvQk · 2024-11-24
>
> Dear Authors
>
> I appreciate your reply. The explanation helps me better understand the paper. I am so sorry I still think a regularization term is incremental as the main contribution.
>
> However, the authors have nice experiments in scRNA-seq. Some theoretical analyses when this tool is applied to negative binomial distributions are also provided. I am sorry those are topics I am not familiar with. I will increase my rating, but keep my confidence low.
>
> I deeply appreciate the authors' paper and the AC's effort.
>
> Best wishes
> reviewer

---

> ### Author Response · Authors · 2024-11-24
> **Many thanks for your reply**
>
> Dear Reviewer tvQk,
>
> We sincerely thank you for the time invested in reviewing our responses and for highlighting the positive aspects of our work. We also appreciate your honesty and openness in considering an increased score despite the low confidence.
>
> Regarding novelty concerns, our regularisation strategy uniquely imposes geometric constraints on a VAE, introducing a novel, unexplored approach. Specifically, we are the first to induce correspondence between geodesics on an (approximately) Euclidean latent space and the statistical manifold spanned by a decoder with arbitrary likelihood. This creates a robust representation space for manifold interpolations across data types with known noise models, an important step forward in scientific domains. This contribution advances trajectory modelling and interpolations in experimental data, addressing a combination of desiderata unmet by current methods like Flow Matching on manifolds or geometry-aware VAEs (see desiderata bullet point list in our rebuttal).
>
> We also encourage you to review the new experimental result directed at **7m6g** (Section I2, Fig. 10). It illustrates how linear latent space paths map to geodesic paths on a 2D circular manifold within a 3D statistical manifold, enabled only by our flattening approach. This example uses a Gaussian likelihood, highlighting the framework’s flexibility beyond the negative binomial case.
>
> We hope these clarifications and additional insights emphasise the novelty and impact of our work, addressing your concerns and encouraging a more favourable evaluation. Thank you again for your thoughtful feedback.
>
> Sincerely,
> The Authors

---

### Official Review · Reviewer_sSiY · 2024-11-06

**Soundness:** 3
**Presentation:** 3
**Contribution:** 3
**Rating:** 6
**Confidence:** 3

**Summary:**

The manuscript is on learning a specific representation tailored to learn dynamics, specifically arising from scRNA-seq. The authors regularize the latent space of a VAE, by the geodesic of a statistical manifold. This way, straight lines in the latent space correspond to geodesics on the statistical manifold. The authors showcase the advantages of the method on toy datasets with known geodesics and scRNA-seq datasets.

**Strengths:**

The presentation of the paper is easy to follow and the method is clearly explained. I like that the authors test their methods on simulated data and real word data. The method is founded on a strong theoretical background.

**Weaknesses:**

In table 1, it is hard to understand the scale of the MSE (Geo-Euc). We understand that $\lambda = 7$ is better than $\lambda=0$, but maybe $8.02$ is still bad. It could be interesting to look at global and local properties that are preserved with the distance matrix. For example, include additional analyses such as Spearman correlation between the ground truth distance matrix and the one from predicted geodesics, or the overlap between k-nearest neighbors of the two distance matrices. This will give a more comprehensive view of how well the geodesic distances are preserved.

**Questions:**

Minor comments and questions;

- In Table 1, consider renaming the 'Model' column to 'Lambda' to more accurately reflect the variable being compared across rows.
- I find Figure 2 a bit hard to interpret, especially the columns a) and b). What should we expect for a Euclidean space? If I understand correctly, both VoR and CN should be uniform on the latent space. It may be beneficial to add these features on a Euclidean space, so that the reader can visually compare the different representations.

- In Figure 2, some points in the FlatVI model still show large VoR and CN values. Could you provide an explanation for why these points persist despite the flattening regularization? Additionally, it would be informative to show examples of geodesics between points from high VoR regions in the FlatVI latent space to demonstrate whether they are indeed straight lines as expected. Consider including a brief discussion of potential reasons for these high VoR and CN points in the FlatVI model.

---

> ### Author Response · Authors · 2024-11-22
>
> We thank the reviewer for their detailed feedback on our work. We appreciate the positive comments regarding the manuscript's content and presentation, and we value their constructive criticism as an opportunity to enhance the quality of our contribution. Below, we provide our responses to the questions and concerns raised during the review process.
>
> > Addition of Spearman correlation and neighbourhood overlap metrics.
>
> We acknowledge the reviewer's comment and have implemented the suggested changes. The updated results are presented in **Tab. 4, section I.1.3**. To compute the metrics of interest, we follow these steps:
>
> - **Sampling:** For each regularisation strength, we randomly sample 50 synthetic observations across 5 repetitions.
> - **Distance Computation:** For all pairs of observations within the sampled batch, we compute both Euclidean and pullback geodesic distances, resulting in two distinct distance matrices. The pullback geodesic approximation is derived using **Eq. 38** from the manuscript.
> - **Neighborhood Evaluation:** The two distance matrices induce neighbourhood structures. We evaluate two metrics:
>   1. **Average Spearman correlation:** We compute the Spearman correlation between Euclidean and geodesic distance vectors for each observation. A high correlation indicates alignment in the ordering of distances from a given observation $i$ across the two metrics.
>   2. **Neighborhood overlap score:** For each observation, we calculate the $k$-nearest neighbours based on both Euclidean and geodesic distance matrices, then compute the average proportion of shared neighbours across all sampled observations.
>
> We report the neighbourhood overlap metric for $k = \{3, 5\}$, reflecting the preservation of local metric properties. Conversely, the Spearman correlation captures the global structure by operating on the entire vector of distances for each observation.
>
> The results in **Tab. 4** demonstrate that regularisation improves both Spearman correlation and neighbourhood overlap values. Specifically, the Spearman correlation starts at relatively high values even before regularisation. However, the neighbourhood overlap score shows more significant improvement with increasing $\lambda$, particularly for $k = 3$, where a regularisation strength of $\lambda$ = 10 yields an average neighbourhood overlap proportion of 0.80. These findings validate our model's principles, where regularisation effectively aligns the pullback geometry closer to Euclidean geometry.
>
> **Notes on geodesic computation**
> 1. We use batches of 50 observations for neighbourhood computations to manage the high computational complexity of estimating geodesics with **Eq. 38**.
> 2. **Eq. 38** provides an approximation of the true geodesic via cubic spline optimisation [1], as the exact geodesic distance in the pullback metric setting is intractable.
>
>
> > Rename the 'Model' column to 'Lambda'.
>
> We applied the recommended change in the updated version of the document.
>
> > Add CN and VoR features to a Euclidean space so that the reader can visually compare the different representations.
>
> Please find the suggested plot in **Fig. 6** of the updated document. The bottom row illustrates the expected behaviour of the VoR and CN metrics in a perfectly Euclidean space. Specifically:
>
> - The **VoR** metric is equal to 0.
> - The **CN** metric is equal to 1.
>
> On the right-hand side, we visualise the geodesic distances between pairs of observations in the Euclidean space. These distances are represented as straight lines connecting the points, similar to the regularised latent space in FlatVI.

---

> > ### Author Response · Authors · 2024-11-22
> >
> > > Could you explain why points with high VoR and CN persist?
> >
> > * We carefully investigated potential reasons for the persistence of points with high VoR and CN values after regularisation in **Appendix I.1.4**. As correctly pointed out by the reviewer, we observed that pairwise geodesics involving points in regions with high VoR and CN still exhibit some curvature, suggesting insufficient flattening. These values are plotted in **Fig. 7** in the Appendix.
> >
> > * In **Fig. 8b**, we overlay the VoR and CN metrics of the standard NB-VAE and its regularised version onto the UMAP plot of the simulated data. This allows us to assess which regions are affected by insufficient flattening. Interestingly, these regions do not correspond between FlatVI and the NB-VAE. For the NB-VAE, a large portion of class 0 regions is impacted by a fast-varying Riemannian metric. **In contrast, for FlatVI, the lack of metric uniformity is more sparse, and CN values are generally lower.** Notably, CN and VoR values are correlated in the latent space of our regularised model (**Fig. 8a**), highlighting well-defined regions where the flattening is imperfect.
> >
> > * In **Fig. 9**, we zoom into observations with high VoR and CN in FlatVI to identify potential patterns. **A common observation is that points with insufficient flattening tend to lie at the intersection of different classes**. This is further supported by the provided boxes showing the label distribution within these subregions. We hypothesise that FlatVI struggles to sufficiently flatten fast-changing regions at these class intersections. In other words, the model has difficulty unfolding distinctive, fast-varying areas of the manifold while maintaining proper reconstruction via the decoder. As a result, the latent manifold loses local Euclideanicity, potentially because the linear assumption might be too restrictive for the level of expressiveness of the decoder in such regions.
> >
> > Together with the figures supporting our hypotheses, we include a comprehensive discussion of our results in **Section I.1.4**.
> >
> >
> > We want to thank the reviewer again for their time and consideration. We hope that our clarifications address all of the raised concerns to the reviewer's satisfaction and look forward to the discussion.
> >
> > [1] Arvanitidis, Georgios, et al. *"Pulling back information geometry."* arXiv preprint arXiv:2106.05367 (2021).

---

> > > ### Comment · Reviewer_sSiY · 2024-11-26
> > >
> > > I would like to thank the authors for answering my questions. I think the new metrics and the additional experiment proposed by reviewer 7m6g help to validate the model.
> > >
> > > I don't have new questions at the moment, but I will keep reading the discussion with the other reviewers.

---

### Author Response · Authors · 2024-11-22
**General comment**

We sincerely thank all reviewers for their valuable feedback, questions, and suggestions. We are particularly encouraged by the positive remarks regarding the presentation of our work, its soundness and solid theoretical foundation, and the thoroughness of our experimental analysis.
While we provide detailed responses to individual comments, we would like to highlight the key ways in which the feedback has enhanced our manuscript:

- As encouraged by reviewers tvQk and L8EZ, we have elaborated on our contribution and restructured the introduction. We believe that the updated manuscript better contextualises the limitations of existing works and highlights key aspects of FLatVI, that enable the application to real-world data, such as scRNA-seq.

- Based on the feedback from reviewer sSiY, we have extended our analysis regarding the Euclideanity of the learned latent space of FlatVI. We make comparisons to Euclidean space and investigate regions with imperfect flattening. We provide a comprehensive discussion of our results in Section I.1.4 of the updated manuscript.

- The feedback from reviewer 7m6g prompted us to evaluate FlatVI in a lighter setting to prove that our regularisation serves its purpose. To this end, we simulated data points sampled from a 2D circular manifold embedded within a 3D Gaussian statistical manifold. Our results, shown in Fig.8, highlight the effectiveness of our approach, with the empirical findings providing support for our model assumptions and design.

Changes to the manuscript are marked in blue.

In summary, we greatly appreciate the constructive feedback we received. The reviewer’s suggestions have strengthened the claims of our work and improved the overall quality of the manuscript.

---

### Comment · Area_Chair_v16G · 2024-11-26

Dear Reviewers sSiY, L8EZ,
If not already, could you please take a look at the authors' rebuttal? Thank you for this important service.
-AC

---

### Author Response · Authors · 2024-12-03
**Summary of Discussion Outcomes and Manuscript Improvements**

Dear AC and reviewers,

We would like to thank you all for your time and support during the review process. The concerns raised by the reviewers have significantly contributed to improving our paper with expanded insights and analysis. Here, we provide a summary of the key clarifications and updates made in response to the reviewers' comments:

- **Reviewer sSiY:**
  We addressed **sSiY**'s concerns by improving the presentation of results in Tab. 1 and adding examples of how Riemannian metrics behave in Euclidean space. We also quantified neighbourhood overlap and Spearman correlation metrics between Euclidean and geodesic distances (Tab. 4), emphasising the value of our regularisation approach. Finally, we analysed cases of sub-optimal flattening performance, offering explanations based on simulated scRNA-seq data. The reviewer acknowledged these updates and expressed no further concerns.

- **Reviewer tvQk:**
  To address **tvQk**'s concerns, we elaborated on the biological and technical significance of our contribution and outlined desiderata uniquely met by our model compared to state-of-the-art generative methods. Following our response, **tvQk** increased their initial score.

- **Reviewer L8EZ:**
  We rephrased parts of our introduction to better position our contribution and clarified that our model regularises towards latent Euclidean geometry rather than fully enforcing it. We also elaborated on computational complexity and clarified parallels with existing geometry-aware representation learning methods. The reviewer acknowledged these updates and expressed a willingness to update their score in coordination with others.

- **Reviewer 7m6g:**
  We clarified the statistical manifold our model operates on, specifically the negative binomial manifold spanned by a VAE decoder, and demonstrated regularisation principles using a simpler simulation with a circular manifold in a 3D Gaussian manifold. Additionally, we corrected notations and claims and extended discussions on runtime and data dimensionality. The reviewer noted satisfaction with most formal aspects while expressing ongoing scepticism and low confidence in assessing the application component.


In addition to the summary, we would like to offer a final remark addressing the primary objective of our contribution. In line with the selected submission track, our FlatVI model integrates elements of information geometry into a predominantly empirical study aimed at improving standard practices for representation learning of cellular data. While ensuring mathematical rigour consistent with existing lines of work, our main objective is to provide the community with a framework that justifies widely-used latent Euclidean operations without compromising the decoding potential or the representation learning capacity of the model.

In other words, what might appear as a simple regularisation is, in fact, a novel and unexplored formulation for simpler and more intuitive high-dimensional cellular manifold interpolations, which is currently **not addressed** by existing methods. We hope the reviewers can recognise the value of our contribution to the single-cell community.


Once again, we are very grateful for the ongoing support.

Best regards,

The authors

---

### Meta-Review · Area_Chair_v16G · 2024-12-19

**Metareview:**

The paper proposes a regularisation technique for variational autoencoders, such that straight lines in the latent domain approximately correspond to geodesic interpolations in the decoded space. Both reviewers and I agree it is an interesting idea, but there were unsolved concerns about what are the precise setup and contribution, and preferences to seeing more theoretical justification and more elaborated details of the empirical advantages. I encourage the authors to consider the reviewers' comments and submit again.

**Additional Comments On Reviewer Discussion:**

Both reviewers and I agree it is an interesting idea, but there were unsolved concerns about what are the precise setup and contribution, and preferences to seeing more theoretical justification and more elaborated details of the empirical advantages.

---

### Decision · Program_Chairs · 2025-01-22

Reject